# Robustness of conditional GANs to noisy labels

**Kiran Koshy Thekumparampil[†], Ashish Khetan[†], Zinan Lin[‡], Sewoong Oh[†]**
[†]University of Illinois at Urbana-Champaign, [‡]Carnegie Mellon University

## Abstract

We study the problem of learning conditional generators from noisy labeled samples, where the labels are corrupted by random noise. A standard training of conditional GANs will not only produce samples with wrong labels, but also generate poor quality samples. We consider two scenarios, depending on whether the noise model is known or not. When the distribution of the noise is known, we introduce a novel architecture which we call Robust Conditional GAN (RCGAN). The main idea is to corrupt the label of the generated sample before feeding to the adversarial discriminator, forcing the generator to produce samples with clean labels. This approach of passing through a matching noisy channel is justified by accompanying multiplicative approximation bounds between the loss of the RCGAN and the distance between the clean real distribution and the generator distribution. This shows that the proposed approach is robust, when used with a carefully chosen discriminator architecture, known as projection discriminator. When the distribution of the noise is not known, we provide an extension of our architecture, which we call RCGAN-U, that learns the noise model simultaneously while training the generator. We show experimentally on MNIST and CIFAR-10 datasets that both the approaches consistently improve upon baseline approaches, and RCGAN-U closely matches the performance of RCGAN.

## 1 Introduction

Conditional generative adversarial networks (GAN) have been widely successful in several applications including improving image quality, semi-supervised learning, reinforcement learning, category transformation, style transfer, image de-noising, compression, in-painting, and super-resolution [30, 13, 49, 36, 26, 58]. The goal of training a conditional GAN is to generate samples from distributions satisfying certain *conditioning* on some correlated features. Concretely, given samples from joint distribution of a data point $x$ and a label $y$, we want to learn to generate samples from the true conditional distribution of the real data $P_{X|Y}$. A canonical conditional GAN studied in literature is the case of discrete label $y$ [30, 36, 35, 32]. Significant progresses have been made in this setting, which are typically evaluated on the quality of the conditional samples. These include measuring inception scores and intra Fréchet inception distances, visual inspection on downstream tasks such as category morphing and super resolution [32], and faithfulness of the samples as measured by how accurately we can infer the class that generated the sample [36].

We study the problem of training conditional GANs with noisy discrete labels. By noisy labels, we refer to a setting where the label $y$ for each example in the training set is randomly corrupted. Such noise can result from an adversary deliberately corrupting the data [7] or from human errors in crowdsourced label collection [12, 18]. This can be modeled as a random process, where a clean data

---

Author emails are thekump2@illinois.edu, ashish.khetan09@gmail.com, zinanl@andrew.cmu.edu, and swoh@illinois.edu. This work used the Extreme Science and Engineering Discovery Environment (XSEDE), which is supported by National Science Foundation grant number OCI-1053575. Specifically, it used the Bridges system, which is supported by NSF award number ACI-1445606, at the Pittsburgh Supercomputing Center (PSC).

point $x \in \mathcal{X}$ and its label $y \in [m]$ are drawn from a joint distribution $P_{X,Y}$ with $m$ classes. For each data point, the label is corrupted by passing through a noisy channel represented by a row-stochastic *confusion matrix* $C \in \mathbb{R}^{m \times m}$ defined as $C_{ij} \triangleq \mathbb{P}(\widetilde{Y} = j | Y = i)$. This defines a joint distribution for the data point $x$ and a noisy label $\widetilde{y}$: $\widetilde{P}_{X,\widetilde{Y}}$. If we train a standard conditional GAN on noisy samples, then it solves the following optimization:

$$\min_{G \in \mathcal{G}} \max_{D \in \mathcal{F}} V(G, D) = \mathop{\mathbb{E}}_{(x,\widetilde{y}) \sim \widetilde{P}_{X,\widetilde{Y}}} [\phi(D(x, \widetilde{y}))] + \mathop{\mathbb{E}}_{z \sim N, y \sim \widetilde{P}_{\widetilde{Y}}} [\phi(1 - D(G(z; y), y))], \quad (1)$$

where $\phi$ is a function of choice, $D$ and $G$ are the discriminator and the generator respectively optimized over function classes $\mathcal{G}$ and $\mathcal{F}$ of our choice, and $N$ is the distribution of the latent random vector. For typical choices of $\phi$, for example $\log(\cdot)$, and large enough function classes $\mathcal{G}$ and $\mathcal{F}$, the optimal conditional generator learns to generate samples from $\widetilde{P}_{X|\widetilde{Y}}$, the corrupted conditional distribution. In other words, it generates samples $X$ from classes other than what it is conditioned on. As the learned distribution exhibits such a bias, we call this naive approach the *Biased GAN*. Under this setting, there is a fundamental question of interest: can we design a novel conditional GAN that can generate samples from the true conditional distribution $P_{X|Y}$, even when trained on noisy samples?

Several aspects of this problem make it challenging and interesting. First, the performance of such robust GAN should depend on how noisy the channel $C$ is. If $C$ is rank-deficient, for instance, then there are multiple distributions that result in the same distribution after the corruption, and hence no reliable learning of the true distribution is possible. We would ideally want a theoretical guarantee that shows such trade-off between $C$ and the robustness of GANs. Next, when the noise is from errors in crowdsourced labels, we might have some access to the confusion matrix $C$ from historical data. On other cases of adversarial corruption, we might not have any information of $C$. We want to provide robust solutions to both. Finally, an important practical challenge in this setting is to correct the noisy labels in the training data. We address all such variations in our approaches and make the following contributions.

**Our contributions.** We introduce two architectures to train conditional GANs with noisy samples.

First, when we have the knowledge of the confusion matrix $C$, we propose RCGAN (Robust Conditional GAN) in Section 2. We first prove that minimizing the RCGAN loss provably recovers the clean distribution $P_{X|Y}$ (Theorem 2), under certain conditions on the class $\mathcal{F}$ of discriminators we optimize over (Assumption 1). We show that such a condition on $\mathcal{F}$ is also necessary, as without it, the training loss can be arbitrarily small while the generated distribution can be far from the real (Theorem 4). The assumption leads to our particular choice of the discriminator in RCGAN, called *projection discriminator* [32] that satisfies all the conditions (Remark 1). Finally, we provide a finite sample generalization bound showing that the loss minimized in training RCGAN does generalize, and results in the learned distribution being close to the clean conditional distribution $P_{X|Y}$ (Theorem 3). Experimental results in benchmark datasets confirm that RCGAN is robust against noisy samples, and improves significantly over the naive Biased GAN.

Secondly, when we do not have access to $C$, we propose RCGAN-U (RCGAN with Unknown noise distribution) in Section 4. We provide experimental results showing that performance gains similar to that of RCGAN can be achieved. Finally, we showcase the practical use of thus learned conditional GANs, by using it to fix the noisy labels in the training data. Numerical experiments confirm that the RCGAN framework provides a more robust approach to correcting the noisy labels, compared to the state-of-the-art methods that rely only on discriminators.

**Related work.** Two popular training methods for generative models are variational auto-encoders [22] and adversarial training [14]. The adversarial training approach has made significant advances in several applications of practical interest. [37, 2, 5] propose new architectures that significantly improve the training in practical image datasets. [58, 16] propose new architectures to transfer the style of one image to the other domain. [26, 43] show how to enhance a given image with learned generator, by enhancing the resolution or making it more realistic. [27, 50] show how to generate videos and [51, 1] demonstrate that 3-dimensional models can be generated from adversarial training. [23] proposes a new architecture encoding causal structures in conditional GANs. [42] introduces the state-of-the-art conditional independence tester. On a different direction, several recent approaches showcase how the manifold learned by the adversarial training can be used to solve inverse problems [9, 57, 53].

Conditional GANs have been proposed as a successful tool for various applications, including class conditional image generation [36], image to image translation [21], and image generation from text [38, 55]. Most of the conditional GANs incorporate the class information by naively concatenating it to the input or feature vector at some middle layer [30, 13, 38, 55]. AC-GANs [36] creates an auxiliary classifier to incorporate class information. Projection discriminator GAN [32] takes an inner product between the embedded class vector and the feature vector. A recent work [31] which proposes spectral normalization shows that high quality image generation on 1000-class ILSVRC2012 dataset [39] can be achieved using projection conditional discriminator.

Robustness of (unconditional) GANs against adversarial or random noise has recently been studied in [10, 52]. [52] studies an adversarial attack that perturbs the discriminator output. The proposed architecture of RCGAN is inspired by a closely related work of AmbientGAN in [10]. AmbientGAN is a general framework addressing any corruption on the image itself (not necessarily just the labels). Given corrupted samples with a known corruption, AmbientGAN applies that corruption to the output of the generator before feeding it to the discriminator. Motivated by the success of AmbientGAN in de-noising, we propose RCGAN. An important distinction is that we make specific architectural choices guided by our theoretical analysis that gives a significant gain in practice (Appendix J). Under the scenario of interest with noisy labels, we provide sharp analyses for both the population loss and the finite sample loss. Such sharp characterizations do not exist for the more general AmbientGAN scenarios. Further, our RCGAN-U does not require the knowledge of the confusion matrix, departing from the AmbientGAN approach. Learning classifiers from noisy labels is a closely related problem. Recently [34, 20] proposed a theoretically motivated classifier which minimizes the modified loss in presence of noisy labels and showed improvement over the robust classifiers [29, 45, 46]. [47] proposed adding noise to the classifier output to match the noise distribution.

**Notation.** For a vector, $\|x\|_p = (\sum_i |x_i|^p)^{1/p}$ is the $\ell_p$-norm. For a matrix, let $\|\|A\|\|_p = \max_{\|x\|_p=1} \|Ax\|_p$ denote the operator norm. Then $\|\|A\|\|_\infty = \max_i \sum_j |A_{ij}|$, $\|\|A\|\|_1 = \max_j \sum_i |A_{ij}|$ and $\|\|A\|\|_2 = \sigma_{\max}(A)$, the maximum singular value. $\mathbb{1}$ is all ones vector and $\mathbf{I}$ is identity matrix. $[n] = \{1, \ldots, n\}$. For a vector $x \in \mathbb{R}^n$, $x_i$ ($i \in [n]$) is its $i$-th coordinate.

## 2 Our first architecture: RCGAN

Training a conditional GAN with noisy samples results in a biased generator. We propose Robust Conditional GAN (RCGAN) architecture which has the following pre-processing, discriminator update, and generator update steps. We assume in this section that the confusions matrix $C$ is known (and the marginal $P_Y$ can easily be inferred), and address the case of unknown $C$ in Section 4.

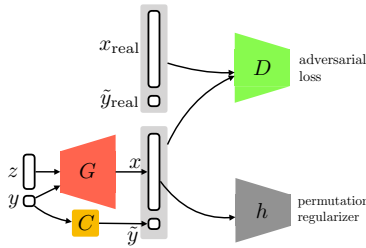

Figure 1: The output $x$ of the conditional generator $G$ is paired with a noisy label $\widetilde{y}$ corrupted by the channel $C$. The discriminator $D$ estimates whether a given labeled sample is coming from the real data $(x_{\text{real}}, \tilde{y}_{\text{real}})$ or generated data $(x, \tilde{y})$. The permutation regularizer $h$ is pre-trained on real data.

**Pre-processing:** We train a classifier $h^*$ to predict the noisy label $\widetilde{y}$ given $x$ under a loss $l$, trained on $h^* \in \arg\min_{h \in \mathcal{H}} \mathbb{E}_{(x,\widetilde{y}) \sim \widetilde{P}_{X,\widetilde{Y}}}[\ell(h(x), \widetilde{y})]$, where $\mathcal{H}$ is a parametric family of classifiers (typically neural networks) and $\widetilde{P}_{X,\widetilde{Y}}$ is the joint distribution of real $x$ and corresponding real noisy $\widetilde{y}$.

**D-step:** We train on the following adversarial loss. In the second term below, $y$ is generated according to $P_Y$ and corresponding noisy labels are generated by corrupting the $y$ according to the conditional distribution $C_y$ which is the $y$-th row of the confusion matrix (assumed to be known):

$$\max_{D \in \mathcal{F}} \mathbb{E}_{(x,\widetilde{y}) \sim \widetilde{P}_{X,\widetilde{Y}}} [\phi(D(x,\widetilde{y}))] + \mathbb{E}_{\substack{z \sim N, y \sim P_Y \\ \widetilde{y}|y \sim C_y}} [\phi(1 - D(G(z;y),\widetilde{y}))] ,$$

where $P_Y$ is the true marginal distribution of the labels, $N$ is the distribution of the latent random vector, and $\mathcal{F}$ is a family of discriminators.

**G-step:** We train on the following loss with some $\lambda > 0$:

$$\min_{G \in \mathcal{G}} \; \mathbb{E}_{\substack{z \sim N, \, y \sim P_Y \\ \widetilde{y}|y \sim C_y}} \left[ \phi \left( 1 - D(G(z; y), \widetilde{y}) \right) + \lambda \, \ell(h^*(G(z; y)), y) \right] , \tag{2}$$

where $\mathcal{G}$ is a family of generators. The idea of using auxiliary classifiers have been used to improve the quality of the image and stability of the training, for example in auxiliary classifier GAN (AC-GAN) [36], and improve the quality of clustering in the latent space [33]. We propose an auxiliary classifiers $h$, mitigating a *permutation error*, which we empirically identified on naive implementation of our idea with no regularizers.

**Permutation regularizer** (controlled by $\lambda$). Permutation error occurs if, when asked to produce samples from a target class, the trained generator produces samples dominantly from a single class but different from the target class. We propose a regularizer $h^*$, which predicts the *noisy* label $\widetilde{y}$. As long as the confusion matrix is diagonally dominant, which is a necessary condition for identifiability, this regularizer encourages the correct permutation of the labels. More regularizers could potentially provide additonal robustness and we discuss one such regularizer (similar to the InfoGAN loss [11]) in Appendix K.

**Theoretical motivation for RCGAN.** When $\lambda = 0$, we get the standard conditional GAN update steps, albeit one which tries to minimize discriminator loss between the noisy real distribution $\widetilde{P}$ and the distribution $\widetilde{Q}$ of the generator when the label is passed through the same noisy channel parameterized by $C$. The main idea of RCGAN is to minimize a certain divergence between noisy real data and noisy generated data. For example, the choice of bounded functions $\mathcal{F} = \{ D : \mathcal{X} \times [m] \to [0,1] \}$ and identity map $\phi(a) = a$ leads to a total variation minimization; The loss minimized in the G-step is the total variation $d_{\text{TV}}(\widetilde{P}, \widetilde{Q}) \triangleq \sup_{S \in \mathcal{X} \times [m]} \{ \widetilde{P}(S) - \widetilde{Q}(S) \}$ between the two distributions with corrupted labels, up to some scaling and some shift. If we choose $\mathcal{F} = \{ D : \mathcal{X} \times [m] \to [0,1] \}$ and $\phi(a) = \log(a)$, then we are minimizing the Jensen-Shannon divergence $d_{\text{JS}}(\widetilde{P}, \widetilde{Q}) \triangleq (1/2) d_{\text{KL}}(\widetilde{P} \| (\widetilde{P} + \widetilde{Q})/2) + (1/2) d_{\text{KL}}(\widetilde{Q} \| (\widetilde{P} + \widetilde{Q})/2)$, where $d_{\text{KL}}(\cdot \| \cdot)$ denotes the Kullback-Leibler divergence. The following theorem provides approximation guarantees for some common divergence measures over noisy channel, justifying our proposed practical approach. We refer to Appendix B for a proof.

**Theorem 1.** *Let $P_{X,Y}$ and $Q_{X,Y}$ be two distributions on $\mathcal{X} \times [m]$. Let $\widetilde{P}_{X,\widetilde{Y}}, \widetilde{Q}_{X,\widetilde{Y}}$ be the corresponding distributions when samples from $P, Q$ are passed through the noisy channel given by the confusion matrix $C \in \mathbb{R}^{m \times m}$ (as defined in Section 1). If $C$ is full-rank, we get,*

$$d_{\text{TV}}\left(\widetilde{P}, \widetilde{Q}\right) \;\leq\; d_{\text{TV}}\left(P, Q\right) \;\leq\; \|\!|\!| C^{-1} |\!|\!\|_\infty \, d_{\text{TV}}\left(\widetilde{P}, \widetilde{Q}\right) , \text{ and} \tag{3}$$

$$d_{\text{JS}}\left(\widetilde{P} \,\Big\|\, \widetilde{Q}\right) \;\leq\; d_{\text{JS}}(P \,\|\, Q) \;\leq\; \|\!|\!| C^{-1} |\!|\!\|_\infty \sqrt{8 \, d_{\text{JS}}\left(\widetilde{P} \,\Big\|\, \widetilde{Q}\right)} . \tag{4}$$

To interpret this theorem, let $Q$ denote the distribution of the generator. The theorem implies that when the noisy generator distribution $\widetilde{Q}$ becomes close to the noisy real distribution $\widetilde{P}$ in total variation or in Jensen-Shannon divergence, then the generator distribution $Q$ must be close to the distribution of real data $P$ in the same metric. This justifies the use of the proposed architecture RCGAN. In practice, we minimize the sample divergence of the two distributions, instead of the population divergence as analyzed in the above theorem. However, these standard divergences are known to not generalize in training GANs [3]. To this end, we provide in Section 3 analyses on *neural network distances*, which are known to generalize, and provide finite sample bounds.

## 3 Theoretical Analysis of RCGAN

It was shown in [3] that standard GAN losses of Jensen-Shannon divergence and Wasserstein distance both fail to generalize with a finite number of samples. On the other hand, more recent advances in analyzing GANs in [56, 6, 4] show promising generalization bounds by either assuming Lipschitz conditions on the generator model or by restricting the analysis to certain classes of distributions. Under those assumptions, where JS divergence generalizes, Theorem 1 justifies the use of the

proposed RCGAN. However, those require the distribution to be Gaussian, mixture of Gaussians, or output of a neural network generator, for example in [4].

In this section, we provide analyses of RCGAN on a distance that generalizes without any assumptions on the distribution of the real data as proven in [3]: *neural network distance*. Formally, consider a class of real-valued functions $\mathcal{F}$ and a function $\phi : [0, 1] \rightarrow \mathbb{R}$ which is either convex or concave. The neural network distance is defined as

$$d_{\mathcal{F},\phi}(P,Q) \quad \triangleq \quad \sup_{D \in \mathcal{F}} \ \underset{(x,y) \sim P}{\mathbb{E}} \left[ \phi \left( D(x,y) \right) \right] + \underset{(x,y) \sim Q}{\mathbb{E}} \left[ \phi \left( 1 - D(x,y) \right) \right] - \mu_\phi . \tag{5}$$

where $P$ is the distribution of the real data, $Q$ is that of the generated data, and $\mu_\phi$ is the constant correction term to ensure that $d_{\mathcal{F},\phi}(P,P) = 0$. We further assume that $\mathcal{F}$ includes three constant functions $D(x,y) = 0$, $D(x,y) = 1/2$, and $D(x,y) = 1$, in order to ensure that $d_{\mathcal{F},\phi}(P,Q) \geq 0$ and $d_{\mathcal{F},\phi}(P,P) = 0$, as shown in Lemma 1 in the Appendix.

The proposed RCGAN with $\lambda = 0$ approximately minimizes the neural network distance $d_{\mathcal{F},\phi}(\widetilde{P}, \widetilde{Q})$ between the two corrupted distributions. In practice, $\mathcal{F}$ is a parametric family of functions from a specific neural network architecture that the designer has chosen. In theory, we aim to identify how the choice of class $\mathcal{F}$ provides the desired approximation bounds similar to those in Theorem 1, but for neural network distances. This analysis leads to the choice of *projection discriminator* [32] to be used in RCGAN (Remark 1). On the other hand, we show in Theorem 4 that an inappropriate choice of the discriminator architecture can cause non-approximation. Further, we provide the sample complexity of the approximation bounds in Theorem 3.

We refer to the un-regularized version with $\lambda = 0$ as simply RCGAN. In this section, we focus on a class of loss functions called Integral Probability Metrics (IPM) where $\phi(x) = x$ [44]. This is a popular choice of loss in GANs in practice [48, 2, 8] and in analyses [4]. We write the induced neural network distance as $d_{\mathcal{F}}(P,Q)$, dropping the $\phi$ in the notation.

## 3.1 Approximation bounds for neural network distances

We define an operation $\circ$ over a matrix $T \in \mathbb{R}^{m \times m}$ and a class $\mathcal{F}$ of functions on $\mathcal{X} \times [m] \rightarrow \mathbb{R}$ as

$$T \circ \mathcal{F} \quad \triangleq \quad \left\{ g(x,y) = \sum_{\widetilde{y} \in [m]} T_{y\widetilde{y}} f(x,\widetilde{y}) \mid f \in \mathcal{F} \right\} . \tag{6}$$

This makes it convenient to represent the neural network distance corrupted by noise with a confusion matrix $C \in \mathbb{R}^{m \times m}$, where $C_{y\widetilde{y}}$ is the probability a label $y$ is corrupted as $\widetilde{y}$. Formally, it follows from (5) and (6) that $d_{\mathcal{F}}(\widetilde{P}, \widetilde{Q}) = d_{C \circ \mathcal{F}}(P,Q)$. We refer to Appendix F for a proof. For $d_{\mathcal{F}}(\widetilde{P}, \widetilde{Q})$ to be a good approximation of $d_{\mathcal{F}}(P,Q)$, we show that the following condition is sufficient.

**Assumption 1.** *We assume that the class of discriminator functions $\mathcal{F}$ can be decomposed into three parts $\mathcal{F} = \{f_1 + f_2 + c \mid f_1 \in \mathcal{F}_1, f_2 \in \mathcal{F}_2\}$ such that $c \in \mathbb{R}$ is any constant and*

- $\mathcal{F}_1$ *satisfies the* inclusion condition*:*

$$T \circ \mathcal{F}_1 \quad \subseteq \quad \mathcal{F}_1 , \tag{7}$$

  *for all $\|T\|_\infty \triangleq \max_i \sum_j |T_{ij}| = 1$; and*

- $\mathcal{F}_2$ *satisfies the* label invariance condition*: there exists a class $\mathcal{F}_2'$ of functions over only $x$, such that*

$$\mathcal{F}_2 \quad = \quad \left\{ \alpha \, g(x,y) \mid g(x,y) = f(x), \text{for any } f(x) \in \mathcal{F}_2', \text{ and } \alpha \in [0,1] \right\} . \tag{8}$$

We discuss the necessity and practical implications of this assumption in Section 3.2, and give examples satisfying these assumptions in Remark 1 and Appendix C. Notice that a trivial class with a single constant zero function satisfies both inclusion and label invariance conditions. For example, we can choose $c = 0$ and also choose to set either $\mathcal{F}_1 = \{f(x,y) = 0\}$ or $\mathcal{F}_2 = \{f(x,y) = 0\}$, in which case $\mathcal{F}$ only needs to satisfy either one of the conditions in Assumption 1. The flexibility that we gain by allowing the set addition $\mathcal{F}_1 + \mathcal{F}_2$ is critical in applying these conditions to practical discriminators, especially in proving Remark 1. Note that in the inclusion condition in Eq. 7, we

require the condition to hold for all max-norm bounded set: $\{T : \max_i \sum_j |T_{ij}| = 1\}$. The reason a weaker condition of all row-stochastic matrices, $\{T : \sum_j T_{ij} = 1\}$, does not suffice is that in order to prove the upper bound in Eq. 9, we need to apply the invariance condition to $\|C^{-1}\|_\infty^{-1} C^{-1} \circ \mathcal{F}$. This matrix $\|C^{-1}\|_\infty^{-1} C^{-1}$ is not row-stochastic, but still max-norm bounded.

We first show that Assumption 1 is sufficient for approximability of the neural network distance from corrupted samples. For two distributions $P_{X,Y}$ and $Q_{X,Y}$ on $\mathcal{X} \times [m]$, let $\widetilde{P}_{X,\widetilde{Y}}$ and $\widetilde{Q}_{X,\widetilde{Y}}$ be the corresponding corrupted distributions respectively, where the label $Y$ is passed through the noisy channel defined by the confusion matrix $C \in \mathbb{R}^{m \times m}$, i.e. $\widetilde{P}(x, \widetilde{y}) = \sum_y P(x,y) C_{y,\widetilde{y}}$.

**Theorem 2.** *If a class of functions $\mathcal{F}$ satisfies Assumption 1, then*

$$d_{\mathcal{F}}(\widetilde{P}, \widetilde{Q}) \;\leq\; d_{\mathcal{F}}(P, Q) \;\leq\; \|C^{-1}\|_\infty d_{\mathcal{F}}(\widetilde{P}, \widetilde{Q}) \;, \tag{9}$$

*where we follow the convention that $\|C^{-1}\|_\infty = \infty$ if $C$ is not full rank.*

We refer to Appendix F for a proof. This gives a sharp characterization on how two distances are related: the one we can minimize in training RCGAN (i.e. $d_{\mathcal{F}}(\widetilde{P}, \widetilde{Q})$) and the true measure of closeness (i.e. $d_{\mathcal{F}}(P, Q)$). Although the latter cannot be directly evaluated or minimized, RCGAN is approximately minimizing the true neural network distance $d_{\mathcal{F}}(P, Q)$ as desired.

The lower bound proves a special case of the data-processing inequality. Two random variables from $P$ and $Q$ get closer in neural network distance, when passed through a stochastic transformation. The upper bound puts a limit on how much closer $\widetilde{P}$ and $\widetilde{Q}$ can get, depending on the noise level. This fundamental trade-off is captured by $\|C^{-1}\|_\infty$. Under the noiseless case where $C$ is the identity matrix, we have $\|C^{-1}\|_\infty = 1$ and we recover a trivial fact that the two distances are equal. On the other extreme, if $C$ is rank deficient, we use the convention that $\|C^{-1}\|_\infty = \infty$ and the two distances can be arbitrarily different. The approximation factor of $\|C^{-1}\|_\infty$ captures how much the space $\mathcal{F}$ can shrink by the noise $C$. This coincides with Theorem 1, where a similar trade-off was identified for the TV distance. In Remark 3 in Appendix D, we show that these bounds cannot be tightened for general $P$, $Q$, and $\mathcal{F}$.

Theorem 2 shows that $(i)$ RCGAN can learn the true conditional distribution, justifying its use; and $(ii)$ performance of RCGAN is determined by how noisy the samples are via $\|C^{-1}\|_\infty$. There are still two loose ends. First, does practical implementation of RCGAN architecture satisfy the inclusion and/or label invariance assumptions? Secondly, in practice we cannot minimize $d_{\mathcal{F}}(\widetilde{P}, \widetilde{Q})$ as we only have a finite number of samples. How much do we lose in this finite sample regime? We give precise answers to each question in the following two sections.

### 3.2 Inclusion and label invariance assumptions

For RCGAN, we propose a popular state-of-the-art discriminator for conditional GANs known as the *projection discriminator* [32], parametrized by $V \in \mathbb{R}^{m \times d_V}$, $v \in \mathbb{R}^{d_v}$, and $\theta \in \mathbb{R}^{d_\theta}$:

$$D_{V,v,\theta}(x, y) \;=\; \mathrm{vec}(y)^T V \psi(x; \theta) + v^T \psi'(x; \theta) \;, \tag{10}$$

where $\psi(x; \theta) \in \mathbb{R}^{d_V}$ and $\psi'(x; \theta) \in \mathbb{R}^{d_v}$ are vector valued parametric functions for some integers $d_V$, $d_v$, and $\mathrm{vec}(y)^T = [\mathbb{I}_{y=1}, \ldots, \mathbb{I}_{y=m}]$. The first term satisfies the inclusion condition, as any operation with $T$ can be absorbed into $V$. The second term is label invariant as it does not depend on $y$. This is made precise in the following remark, whose proof is provided in Appendix G. Together with this remark, the approximability result in Theorem 2 justifies the use of projection discriminators in RCGAN, which we use in all our experiments.

**Remark 1.** *The class of projection discriminators $\{D_{V,v,\theta}(x, y)\}_{V \in \mathcal{V}_1, v \in \mathcal{V}_2, \theta \in \Theta}$ defined in Eq. 10 satisfies Assumption 1 for any $\psi$, $\psi'$, and $\Theta$, if $\mathcal{V}_1 = \left\{ V \in \mathbb{R}^{m \times d_V} \mid \max_i |V_{ij}| \leq 1 \text{ for all } j \in [d_V] \right\}$, and $\mathcal{V}_2 = \left\{ v \in \mathbb{R}^{d_v} \mid \|v\| \leq 1 \right\}$.*

Other choices of $\mathcal{V}_1$ and $\mathcal{V}_2$ are also possible. For example, $\mathcal{V}_1' = \{V \in \mathbb{R}^{m \times d_V} \mid \sum_j \max_i |V_{ij}| \leq 1\}$ or $\mathcal{V}_1'' = \{V \in \mathbb{R}^{m \times d_V} \mid \|V\|_\infty = \max_i \sum_j |V_{ij}| \leq 1\}$ are also sufficient. We find the proposed choice of $\mathcal{V}_1$ easy to implement, as a column-wise $L_\infty$-norm normalization via projected gradient descent. We describe implementation details in Appendix L. In Appendix E, we show that Assumption 1 is also necessary.

### 3.3 Finite sample analysis

In practice, we do not have access to the probability distributions $\widetilde{P}$ and $\widetilde{Q}$. Instead, we observe a set of samples of a finite size $n$, from each of them. In training GAN, we minimize the *empirical neural network distance*, $d_{\mathcal{F}}(\widetilde{P}_n, \widetilde{Q}_n)$, where $\widetilde{P}_n$ and $\widetilde{Q}_n$ denote the empirical distribution of $n$ samples. Inspired from the recent generalization results in [3], we show that this empirical distance minimization leads to small $d_{\mathcal{F}}(P, Q)$ up to an additive error that vanishes with an increasing sample size. As shown in [3], Lipschitz and bounded function classes are critical in achieving sample efficiency for GANs. We follow the same approach over a similar function class. Let

$$\mathcal{F}_{p,L} = \{D_u(x,y) \in [0,1] \mid D_u(x,y) \text{ is } L\text{-Lipschitz in } u \text{ and } u \in \mathcal{U} \subseteq \mathbb{R}^p\}, \tag{11}$$

be a class of bounded functions with parameter $u \in \mathbb{R}^p$. We say that $\mathcal{F}$ is $L$-Lipschitz in $u$ if

$$|D_{u_1}(x,y) - D_{u_2}(x,y)| \leq L\|u_1 - u_2\|, \quad \forall u_1, u_2 \in \mathcal{U}, \ x \in \mathcal{X}, \ y \in [m]. \tag{12}$$

**Theorem 3.** *For any class $\mathcal{F}_{p,L}$ of bounded Lipschitz functions $D_u(x,y)$ satisfying Assumption 1, there exists a universal constant $c > 0$ such that*

$$d_{\mathcal{F}_{p,L}}(\widetilde{P}_n, \widetilde{Q}_n) - \epsilon \ \leq \ d_{\mathcal{F}_{p,L}}(P,Q) \ \leq \ \|C^{-1}\|_{\infty} \left( d_{\mathcal{F}_{p,L}}(\widetilde{P}_n, \widetilde{Q}_n) + \epsilon \right), \tag{13}$$

*with probability at least $1 - e^{-p}$ for any $\varepsilon > 0$ and $n$ large enough, $n \ \geq \ (c\,p\,/\epsilon^2) \log(pL/\epsilon)$ .*

We refer to Appendix I for a proof. This justifies the proposed RCGAN which minimizes $d_{\mathcal{F}}(\widetilde{P}_n, \widetilde{Q}_n)$, as it leads to the generator $Q$ being close to the real distribution $P$ in neural network distance, $d_{\mathcal{F}}(P, Q)$. These bounds inherit the approximability of the population version from Theorem 2.

## 4 Our second architecture: RCGAN-U

In many real world scenarios the confusion matrix $C$ is unknown. We propose RCGAN-Unknown (RCGAN-U) algorithm which jointly estimates the real distribution $P$ and the noise model $C$. The pre-processing and D steps of the RCGAN-U are the same as those of RCGAN, assuming the current guess $M$ of the confusion matrix. As the G-step in (2) is not differentiable in $C$, we use the following reparameterized estimator of the loss, motivated by similar technique in training classifiers from noisy labels:

$$\min_{G \in \mathcal{G}, M \in \mathcal{C}} \ \mathbb{E}_{z \sim N \atop y \sim P_Y} \left[ \phi_M \left( G(z; y), y, D \right) + \lambda l(h^*(G(z; y)), y) \right]$$

where $\mathcal{C}$ is the set of all transition matrices and $\phi_M(x, y, D) = \sum_{\widetilde{y} \in [m]} M_{y\widetilde{y}} \phi(1 - D(x, \widetilde{y}))$.

## 5 Experiments

Implementation details are explained in Appendix L. We consider one-coin based models, which are parameterized by their label accuracy probability $\alpha$. In this model a sample with true label $y$ is flipped uniformly at random to label $\widetilde{y}$ in $[m] \setminus \{y\}$ with probability $1 - \alpha$. The entries of its confusion matrix $C$, will then be $C_{ii} = \alpha$ and $C_{i \neq j} = (1 - \alpha)/(m - 1)$, where $m$ is the number of classes. We call this model *uniform flipping* model. Code to reproduce our experiments is available at `https://github.com/POLane16/Robust-Conditional-GAN`.

**Baselines.** First is the *biased GAN*, which is a conditional GAN applied directly on the noisy data. The loss is hence biased, and the true conditional distribution is not the optimal solution of this biased loss. Next natural baseline is using de-biased classifier as the discriminator, motivated by the approach of [34] on learning classifiers from noisy labels. The main insight is to modify the loss function according to $C$, such that in expectation the loss matches that of the clean data. We refer to this approach as *unbiased GAN*. Concretely, when training the discriminator, we propose the following (modified) de-biased loss:

$$\max_{D \in \mathcal{F}} \ \mathbb{E}_{(x,\widetilde{y}) \sim \widetilde{P}_{X, \widetilde{Y}}} \Big[ \sum_{y \in [m]} (C^{-1})_{\widetilde{y}y} \phi \left( D(x,y) \right) \Big] + \mathbb{E}_{z \sim N \atop y \sim P_Y} \left[ \phi \left( 1 - D(G(z; y), y) \right) \right]. \tag{14}$$

This is unbiased, as the first term is equivalent to $\mathbb{E}_{(x,y) \sim P_{X,Y}}[\phi(D(x,y))]$, which is the standard GAN loss with clean samples. However, such de-biasing is sensitive to the condition number of $C$, and can become numerically unstable for noisy channels as $C^{-1}$ has large entries [20]. For both the dataset, we use linear classifiers for permutation regularizer of the RCGAN-U architecture.

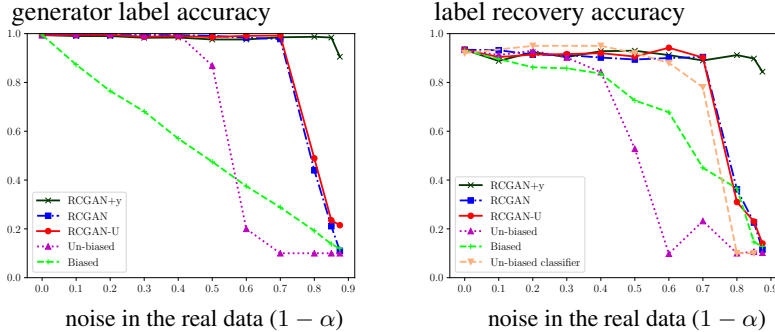

Figure 2: Noisy MNIST dataset: Our RCGAN models consistently improves upon all competing baseline approaches in generator label accuracy (left). The trend continues in label recovery accuracy (right), where our proposed RCGAN-classifiers improves upon *unbiased classifier* [34], which is one of the state-of-the-art approaches tailored for label recovery.

## 5.1 MNIST

We train five architectures on MNIST dataset corrupted by the uniform flipping noise: RCGAN+y, RCGAN, RCGAN-U, unbiased GAN, and biased GAN. RCGAN+y architecture has the same architecture as RCGAN but the input to the first layer of its discriminator is concatenated with a one-hot representation of the label. We discuss our techniques to overcome the challenges involved in training RCGAN+y in Appendix L.

Conditional generators can be used to generate samples $x$ from a particular class $y$, in the classes it learned. We then can use a pre-trained classifier $f$ to compare $y$ to the true class of the sample, $f(x)$ (as perceived by the classifier $f$). We compare the *generator label accuracy* defined as $\mathbb{E}_{y \sim P_Y, Z \sim N}[\mathbb{I}_{\{y=f(G(z,y))\}}]$, in Figure 2, left panel. We generated 10k labels chosen uniformly at random and corresponding conditional samples from the generators, and calculated the generator label accuracy using a CNN classifier pre-trained on the clean MNIST data to an accuracy of 99.2%. The proposed RCGAN significantly improves upon the competing baselines, and achieves almost perfect label accuracy until a high noise of $\alpha = 0.3$. RCGAN+y further improves upon RCGAN and to gain very high accuracy even at $\alpha = 0.125$. The high accuracy of RCGAN-U suggests that robust training is possible without prior knowledge of the confusion matrix $C$. As expected, biased GAN has an accuracy of approximately $1 - \alpha$.

An immediate application of robust GANs is recovering the true labels of the noisy training data, which is an important and challenging problem in crowdsourcing. We propose a new meta-algorithm, which we call cGAN-label-recovery, which use any conditional generator $G(z, y)$ trained on the noisy samples, to estimate the true label, as $\hat{y}$, of a sample $x$ using the following optimization.

$$\hat{y} \in \arg\min_{y \in [m]} \left\{ \min_{z_y} \||G(z_y, y) - x\||_2^2 \right\}. \tag{15}$$

In the right panel of Figure 2 we compare the *label recovery accuracy* of the meta-algorithm using the five conditional GANs, on 500 randomly chosen noisy training samples. This is also compared to a state-of-the-art method [34] for label recovery, which proposed minimizing unbiased loss function given the noisy labels and the confusion matrix. This unbiased classifier, was shown to outperforms the robust classifiers [29, 45, 46] and can be used to predict the true label of the training examples. In Figures 5 of Appendix M, we show example images from all the generators.

## 5.2 CIFAR-10

In Figure 3, we show the inception score [40] and the label accuracy of the conditional generator for the four approaches: our proposed RCGAN and RCGAN-U, against the baselines Unbiased (Section 5) and Biased (Section 1) GANs trained using CIFAR-10 images [24], while varying the label accuracy of the real data under uniform flipping model. In RCGAN-U, even with the regularizer, the learned confusion matrix was a permuted version of the true $C$, possibly because a linear classifier might be too simple to classify CIFAR images. To combat this, we initialized the confusion matrix $M$ to be diagonally dominant (Appendix L).

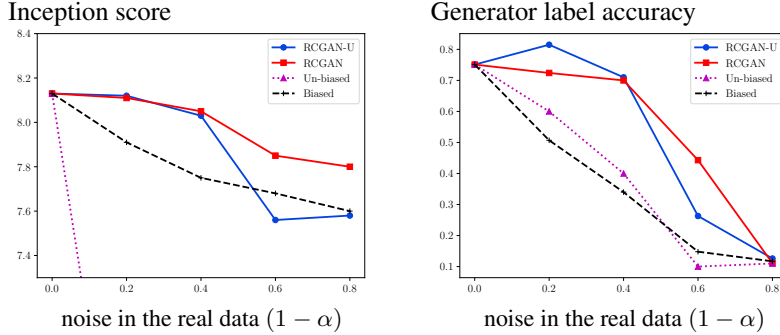

Figure 3: Noisy CIFAR-10 dataset: Our RCGAN (red) and RCGAN-U (blue) consistently improves upon Unbiased (magenta) and Biased (black) GANs trained on noisy CIFAR-10 in inception scores (left) and in generator label accuracy (right).

In the left panel of Figure 3, our RCGAN and RCGAN-U consistently achieve higher inception scores than the other two approaches. The Unbiased GAN is highly unstable and hence produces garbage images for large noise (Fig. 6), possibly due to numerical instability of $\|C^{-1}\|_\infty$, as noted in [20]. This confirms that robust GANs not only produce images from the correct class, but also produce better quality images. In the right panel of Figure 3, we report the generator label accuracy (Section 5.1) on 1k samples generated by each GAN. We classify the generator images using a ResNet-110 model[1] trained to an accuracy of 92.3% on the noiseless CIFAR-10 dataset. Biased GAN has significantly lower label accuracy whereas the Unbiased GAN has low inception score. In Figure 6 in Appendix M, we show example images from the three generators for the different flipping probabilities. We believe that the gain in using the proposed robust GANs will be larger, when we train to higher accuracy with larger networks and extensive hyper parameter tuning, with latest innovations in GAN architectures, for example [54, 28, 17, 19, 41].

## 6   Conclusion

Standard conditional GANs can be sensitive to noise in the labels of the training data. We propose two new architectures to make them robust, one requiring the knowledge of the distribution of the noise and another which does not, and demonstrate the robustness on benchmark datasets of CIFAR-10 and MNIST. We further showcase how the learned generator can be used to recover the corrupted labels in the training data, which can potentially be used in practical applications. The proposed architecture combines the noise adding idea of AmbientGAN [10], projection discriminator of [32], and regularizers similar to those in InfoGAN [11]. Inspired by AmbientGAN [10], the main idea is to pair the generator output image with a label that is passed through a noisy channel, before feeding to the discriminator. We justify this idea of noise adding by identifying a certain class of discriminators that have good generalization properties. In particular, we prove that projection discriminator, introduced in [32], has a good generalization property. We showcase that the proposed architecture, when trained with a regularizer, has superior robustness on benchmark datasets.

### Acknowledgement

This work is supported by NSF awards CNS-1527754, CCF-1553452, CCF-1705007, RI-1815535 and Google Faculty Research Award. This work used the Extreme Science and Engineering Discovery Environment (XSEDE), which is supported by National Science Foundation grant number OCI-1053575. Specifically, it used the Bridges system, which is supported by NSF award number ACI-1445606, at the Pittsburgh Supercomputing Center (PSC). This work is partially supported by the generous research credits on AWS cloud computing resources from Amazon.

## Footnotes

[1]`https://github.com/wenxinxu/resnet-in-tensorflow`

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
