[Supplementary Material]

# Appendix

## A    Notations and Lemmas

### A.1    Additional Notation

Here we define some additional notations required for the proof. We define certain notations before we provide the main theoretical contributions of our paper. If $f(x, y)$ is a function of two variable of $x, y$, where $y \in [m]$, then $\overline{f}(x)$ is the vector $[f(x, 1), \cdots, f(x, m)]^T$. If $P_{X,Y}$ is probability distribution of $(X, Y) \in \mathcal{X} \times [m]$, then $P_{Y|X=x}$ is the conditional distribution of $Y$ given $X = x$.

For a matrix $A$, let $\|A\|_p = \max_{\|x\|_p=1} \|Ax\|_p, \forall p \in \mathbb{N} \cup \{0, \infty\}$. Then $\|A\|_\infty = \max_i \sum_j |A_{ij}|$, $\|A\|_1 = \max_j \sum_i |A_{ij}|$ and $\|A\|_2 = \sigma_{\max}(A)$, the maximum singular value. $\mathbb{1}$ is all ones vector with appropriate dimensions and $\mathbf{I}$ is identity matrix with appropriate dimensions. $[n] = \{1, 2, \ldots, n\}, \forall n \geq 1$. For a vector $x \in \mathbb{R}^n$, $x_i$ ($i \in [n]$) is its $i$-th coordinate.

For the sake of proof we will assume that $\mathcal{F}$ is class of vector functions of the form $D(x) \in \mathbb{R}^m$. In terms of the notation in the main material original $D(x, y)$ is $D(x)_y$ here. For a class $\mathcal{F}$ of vector valued functions $D : \mathcal{X} \to \mathbb{R}^n$. Therefore we re-define the operation $\circ$ between a matrix $T \in \mathbb{R}^{n \times n}$ and $\mathcal{F}$ as,

$$T \circ \mathcal{F} = \{TD(\cdot) \mid f(\cdot) \in \mathcal{F}\}.$$

If $P_{X,Y}$ is probability distribution of $(X, Y) \in \mathcal{X} \times [m]$, then $P_{Y|X=x}$ is the conditional discrete distribution of $Y$ given $X = x$, $p_X(x)$ is the marginal density of $X$, and

$$\overline{P}_{Y|X=x} = [P_{Y|X=x}(Y = 1),\, P_{Y|X=x}(Y = 1),\, \ldots,\, P_{Y|X=x}(Y = m)]^T, \text{ and} \tag{16}$$

$$\overline{p_X}(x) = p_X(x)\overline{P}_{Y|X=x} \tag{17}$$

### A.2    Supporting Lemmas

**Lemma 1** (Characterization of neural network distance). *$d_{\mathcal{F},\phi}(P, Q) \geq 0$ for all $P, Q$. And if $\phi$ is a convex or concave function, then the Neural network distance is $0$ when the distributions are same, i.e. $d_{\mathcal{F},\phi}(P, P) = 0$.*

*Proof.* For concave $\phi(\cdot)$ we define $\mu_\phi = \phi(1/2)$. Since, by definition $D = 1/2\,\mathbb{1}$ is feasible solution to the optimization problem in (5), thus $d_{\mathcal{F},\phi}(P, Q) \geq 0$.

$$
\begin{aligned}
d_{\mathcal{F},\phi}(P, P) &= \sup_{D \in \mathcal{F}} \, \mathop{\mathbb{E}}_{(x,y)\sim P} [\phi(D(x)_y)] + \mathop{\mathbb{E}}_{(x,y)\sim P} [\phi(1 - D(x)_y)] - 2\phi(1/2) \\
&\leq \sup_{D \in \mathcal{F}} \, 2\,\phi\left( \mathop{\mathbb{E}}_{(x,y)\sim P}\left[ \frac{1}{2}(D(x)_y + 1 - D(x)_y) \right] \right) - 2\phi(1/2) \\
&= \sup_{D \in \mathcal{F}} \, 2\,\phi(1/2) - 2\phi(1/2) = 0
\end{aligned}
$$

The inequality in second line follows from Jensen's inequality for concave $\phi(\cdot)$.

For convex $\phi(\cdot)$ we define $\mu_\phi = \phi(0) + \phi(1)$. Since, by definition $D = 1$ is feasible solution to the optimization problem in (5), thus $d_{\mathcal{F},\phi}(P, Q) \geq 0$.

$$
\begin{aligned}
d_{\mathcal{F},\phi}(P, P) &= \sup_{D \in \mathcal{F}} \, \mathop{\mathbb{E}}_{(x,y)\sim P} [\phi(D(x)_y)] + \mathop{\mathbb{E}}_{(x,y)\sim P} [\phi(1 - D(x)_y)] - \phi(0) - \phi(1) \\
&= \sup_{D \in \mathcal{F}} \, \mathop{\mathbb{E}}_{(x,y)\sim P} [\phi(D(x)_y) + \phi(1 - D(x)_y)] - (\phi(0) + \phi(1)) \\
&\leq \sup_{D \in \mathcal{F}} \, \mathop{\mathbb{E}}_{(x,y)\sim P} [\phi(0) + \phi(1)] - (\phi(0) + \phi(1)) = 0
\end{aligned}
$$

The last inequality follows from Jensen's inequality for convex $\phi(\cdot)$ $\qquad\square$

This Lemma 1 ensures that all the multiplicative lower bounds and upper bounds in Theorem 4 and its corollaries implies recoverability.

**Lemma 2.** *If $P$ is a distributions on $\mathcal{X} \times [m]$ and $\widetilde{P}$ is the distribution of sample $(X, \widetilde{Y})$ of $P$ when passed through the noisy-channel given by the confusion matrix $C \in \mathbb{R}^{m \times m}$ (as defined in Section 1). Then,*

$$\overline{\widetilde{P}}_{\widetilde{Y}|X=x} = C^T \overline{P}_{Y|X=x}, \tag{18}$$

*where $\overline{P}_{Y|X=x} = [P_{Y|X=x}(Y=1), P_{Y|X=x}(Y=1), \ldots, P_{Y|X=x}(Y=m)]^T$.*

*Proof.*

$$\widetilde{P}_{\widetilde{Y}|X=x}(\widetilde{Y}=j) = \sum_{i \in [m]} \mathbb{P}\left(\widetilde{Y}=j|Y=i\right) P_{Y|X=x}(Y=j), \forall j \in [m]$$

$$\widetilde{P}_{\widetilde{Y}|X=x}(\widetilde{Y}=j) = \sum_{i \in [m]} C_{ij} P_{Y|X=x}(Y=j), \forall j \in [m]$$

$$\overline{\widetilde{P}}_{\widetilde{Y}|X=x} = C^T \overline{P}_{Y|X=x}$$

$\square$

## B   Proof of Theorem 1

We first prove the approximation bounds for total variation distance in Eq. (3), and then use it to prove similar bounds for the Jensen-Shannon divergence in Eq. (4). Recall that total variation distance between $P$ and $Q$ can be written in several ways:

$$
\begin{aligned}
d_{\mathrm{TV}}(P,Q) &= \frac{1}{2} \max_{\coprod_v S^{(v)} = \mathcal{X}} \sum_{v \in \{-1,1\}^m} \langle v, P(S^{(v)}, \cdot) - Q(S^{(v)}, \cdot) \rangle \\
&= \frac{1}{2} \max_{\coprod_v S^{(v)} = \mathcal{X}} \sum_{v \in \{-1,1\}^m} \sum_{y \in [m]} |P(S^{(v)}, y) - Q(S^{(v)}, y)| \\
&= \frac{1}{2} \max_{\coprod_v S^{(v)} = \mathcal{X}} \sum_{v \in \{-1,1\}^m} \|P(S^{(v)}, \cdot) - Q(S^{(v)}, \cdot)\|_1,
\end{aligned}
$$

where we used the notation of a row-vector $P(S^{(v)}, \cdot) = [P(S^{(v)}, 1), \cdots, P(S^{(v)}, m)]$ and $\coprod$ is the disjoint union of sets. The lower bound on $d_{\mathrm{TV}}(\widetilde{P}, \widetilde{Q})$ follows that

$$
\begin{aligned}
d_{\mathrm{TV}}(P,Q) &= \frac{1}{2} \max_{\coprod_v S^{(v)} = \mathcal{X}} \sum_{v \in \{-1,1\}^m} \langle v, P(S^{(v)}, \cdot) - Q(S^{(v)}, \cdot) \rangle \\
&\overset{(a)}{=} \frac{1}{2} \max_{\coprod_v S^{(v)} = \mathcal{X}} \sum_{v \in \{-1,1\}^m} \langle v, \left(\widetilde{P}(S^{(v)}, \cdot) - \widetilde{Q}(S^{(v)}, \cdot)\right) C^{-1} \rangle \\
&\overset{(b)}{\leq} \|\|C^{-T}\|\|_1 \frac{1}{2} \max_{\coprod_v S^{(v)} = \mathcal{X}} \sum_{v \in \{-1,1\}^m} \|v\|_\infty \|\widetilde{P}(S^{(v)}, \cdot) - \widetilde{Q}(S^{(v)}, \cdot)\|_1 \\
&\overset{(c)}{=} \|\|C^{-T}\|\|_1 \, d_{\mathrm{TV}}(\widetilde{P}, \widetilde{Q})
\end{aligned}
$$

where $(a)$ follows from the fact that $\widetilde{P}(\{S_y\}_{y \in [m]}, \cdot) = P(\{S_y\}_{y \in [m]}, \cdot) C$, $(b)$ follows from the fact that $v^T A x \leq \|v\|_\infty \|Ax\|_1 \leq \|v\|_\infty \|\|A\|\|_1 \|x\|_1$, and $(c)$ follows from $\|\|A\|\|_1 = \|\|A^T\|\|_\infty$ and $\|v\|_\infty = 1$. The upper bound follows from similar arguments:

$$
\begin{aligned}
d_{\mathrm{TV}}\left(\widetilde{P}, \widetilde{Q}\right) &\leq \|\|C^T\|\|_1 \frac{1}{2} \max_{\coprod_v S^{(v)} = \mathcal{X}} \sum_{v \in \{-1,1\}^m} \|v\|_\infty \|P(S^{(v)}, \cdot) - Q(S^{(v)}, \cdot)\|_1 \\
&= d_{\mathrm{TV}}(P,Q)
\end{aligned}
$$

where last equality uses the fact that $\|\|C^T\|\|_1 = 1$ for all row-stochastic matrices $C$.

To prove the approximation bounds for Jensen-Shannon divergence, we use the following lemma that bounds the JS divergence by the TV distance.

**Lemma 3.** $\frac{1}{2} d_{\text{TV}}(P, Q)^2 \leq d_{\text{JS}}(P \parallel Q) \leq 2 d_{\text{TV}}(P, Q)$.

A proof is provided in Section B.1. The following series of inequalities follow from this lemma.

$$d_{\text{JS}}(P \parallel Q) \overset{(a)}{\geq} d_{\text{JS}}\left(\widetilde{P} \,\Big\|\, \widetilde{Q}\right) \overset{(b)}{\geq} \frac{1}{2} d_{\text{TV}}\left(\widetilde{P}, \widetilde{Q}\right)^2$$

$$\overset{(c)}{\geq} \frac{\|\|C^{-1}\|\|_\infty^{-2}}{2} d_{\text{TV}}(P, Q)^2$$

$$\overset{(b)}{\geq} \frac{\|\|C^{-1}\|\|_\infty^{-2}}{8} d_{\text{JS}}(P \parallel Q)^2$$

where $(a)$ is the data-processing inequality for $f$-divergences, $(b)$ uses Lemma 3, and $(c)$ uses equation (3).

## B.1 Proof of Lemma 3

$$d_{\text{KL}}\left(P \,\Big\|\, \frac{P+Q}{2}\right) = \underset{X \sim P}{\mathbb{E}}\left[\log \frac{2P(X)}{P(X) + Q(X)}\right]$$

$$\overset{(a)}{\leq} \underset{X \sim P}{\mathbb{E}}\left[\frac{2P(X) - (P(X) + Q(X))}{P(X) + Q(X)}\right]$$

$$\leq \sup_X \frac{2P(X)}{P(X) + Q(X)} \cdot \underset{X \sim P}{\mathbb{E}}\left[\frac{|2P(X) - (P(X) + Q(X))|}{2P(X)}\right]$$

$$\leq 2 \cdot 2d_{\text{TV}}\left(P, \frac{P+Q}{2}\right)$$

$$\leq 2 d_{\text{TV}}(P, Q) , \tag{19}$$

where $(a)$ uses $\log x \leq x - 1$.

$$d_{\text{JS}}(P \parallel Q) = \frac{1}{2} d_{\text{KL}}\left(P \,\Big\|\, \frac{P+Q}{2}\right) + \frac{1}{2} d_{\text{KL}}\left(Q \,\Big\|\, \frac{P+Q}{2}\right)$$

$$\overset{(b)}{\geq} d_{\text{TV}}\left(P, \frac{P+Q}{2}\right)^2 + d_{\text{TV}}\left(Q, \frac{P+Q}{2}\right)^2$$

$$= \frac{1}{2} d_{\text{TV}}(P, Q)^2$$

where $(a)$ uses equation (19), and $(b)$ uses Pinsker's inequality $0.5 \, d_{\text{KL}}(P \parallel Q) \leq d_{\text{TV}}(P, Q)^2$

## C Examples satisfying Assumption 1

Several classes of functions that are used in practice and studied in theory indeed satisfy our Assumption 1. For example, consider the set of 1-Lipschitz continuous and bounded functions $\mathcal{F} = \{f : \mathbb{R}^d \times [m] \to \mathbb{R} \,|\, 0 \leq f(x, y) \leq 1 \text{ for all } x \text{ and } y, \text{ and } |f(x_1, y_1) - f(x_2, y_2)| \leq \|x_1 - x_2\| + |y_1 - y_2| \text{ for all } x_1, x_2, y_1, \text{ and } y_2\}$, which was studied in [3] in the context of generalization bounds for the neural network distance. It follows that this $\mathcal{F}$ satisfies Assumption 1, with $c = 1/2$ and $\mathcal{F} - 1/2 \in \mathcal{F}_1$. This is a special case of some examples of classes of functions satisfying the assumption, that we provide in the following Remark. A proof is provided in Appendix G.

**Remark 2.** *The following classes of discriminators satisfy the inclusion condition in Assumption 1:*

1. *Class of all bounded functions $D : \mathcal{X} \times [m] \to [c_1, c_2]$ for any $c_1 \leq c_2 \in \mathbb{R}$.*

2. *Class of all bounded functions $D : \mathcal{X} \times [m] \to [c_1, c_2]$, which are L-Lipschitz in x for any $c_1 \leq c_2 \in \mathbb{R}$ and $L \geq 0$.*

3. *Class of all bounded functions $D : \mathcal{X} \times [m] \to [c_1, c_2]$, which are L-Lipschitz in x and y for any $c_1 \leq c_2 \in \mathbb{R}$ and $L \geq (c_2 - c_1)$.*

# D    Tightness of Theorem 2

**Remark 3.** *For any full-rank confusion matrix $C \in \mathbb{R}^{d_1 \times d_2}$, there exist pairs of distributions $(P_1, Q_1)$ and $(P_2, Q_2)$, and a function class $\mathcal{F}$ satisfying Assumption 1, such that*

  *1. $d_\mathcal{F}(\widetilde{P}_1, \widetilde{Q}_1) = d_\mathcal{F}(P_1, Q_1)$, and*

  *2. $d_\mathcal{F}(P_2, Q_2) = |\!|\!|C^{-1}|\!|\!|_\infty d_\mathcal{F}(\widetilde{P}_2, \widetilde{Q}_2)$.*

*Proof.* Let $\mathcal{X} = \{x_1, x_2\}$ and $\mathcal{F} = \{f \mid f(x, y) \in [-1, 1]\}$. From Remark 2, we know that $\mathcal{F}$ satisfies Assumption 1. Then it is easy to check that $d_\mathcal{F}(P, Q) = |\!|\!|(P - Q)((\{x_1\}, \cdot)|\!|\!|_1 + |\!|\!|(P - Q)((\{x_2\}, \cdot)|\!|\!|_1$, where we used the notation $P(\{x\}, \cdot) = [P(\{x\}, 1), \cdots, P(\{x\}, m)]$.

  1. **Lower bound:** It is easy to show that there exists $P_{X,Y}$ and $Q_{X,Y}$ such that $(P - Q)((\{x_1\}, \cdot) = \epsilon \mathbb{1}$ and $(P - Q)((\{x_2\}, \cdot) = -\epsilon \mathbb{1}$ for any $\epsilon \in [0, 1/m]$. Thus,
  $$d_\mathcal{F}(P, Q) = |\!|\!|\epsilon \mathbb{1}|\!|\!|_1 + |\!|\!|-\epsilon \mathbb{1}|\!|\!|_1 = 2m\epsilon. \tag{20}$$
  Then we can show that,
  $$(\widetilde{P} - \widetilde{Q})(\{x\}, \cdot) = C^T (P - Q)(\{x\}, \cdot) = \pm \epsilon\, C^T \mathbb{1} \tag{21}$$
  Thus,
  $$d_\mathcal{F}(\widetilde{P}, \widetilde{Q}) = 2 |\!|\!|\pm \epsilon C^T \mathbb{1}|\!|\!|_1 = 2\epsilon\, \mathbb{1}^T C^T \mathbb{1} = 2\epsilon\, \mathbb{1}^T \mathbb{1} = 2m\epsilon, \tag{22}$$
  where the penultimate equality follow from $C\mathbb{1} = \mathbb{1}$, since $C$ is row-stochastic.

  2. **Upper bound:** We can again show that there exists $P_{X,Y}$ and $Q_{X,Y}$ such that $(P - Q)((\{x_1\}, \cdot) = \epsilon C^{-T} v$ and $(P - Q)((\{x_2\}, \cdot) = -\epsilon C^{-T} v$, where $v \in \arg\max_{v_i \in \pm 1} |\!|\!|C^{-T} v|\!|\!|_1$, for sufficiently small $\epsilon$. Thus,
  $$d_\mathcal{F}(P, Q) = |\!|\!|\pm \epsilon C^{-T} v|\!|\!|_1 = 2\epsilon |\!|\!|C^{-T}|\!|\!|_1 |\!|\!|v|\!|\!|_1 = 2\epsilon |\!|\!|C^{-1}|\!|\!|_\infty |\!|\!|v|\!|\!|_1. \tag{23}$$
  Using similar steps as the above lower-bound case, we can show that,
  $$d_\mathcal{F}(\widetilde{P}, \widetilde{Q}) = 2 |\!|\!|\pm \epsilon C^T C^{-T} v|\!|\!|_1 = 2\epsilon |\!|\!|v|\!|\!|_1. \tag{24}$$

  $\square$

# E    Necessity of Assumption 1

In this section, we ask if Assumption 1 is necessary also. We show that for all pairs of distributions $(P, Q)$ satisfying the following technical conditions, and all confusion matrix $C$, there exists a class $\mathcal{F}$ where approximation bounds in (9) fail.

**Assumption 2.** *We consider a pair of distributions $P_{X,Y}$ and $Q_{X,Y}$ and a confusion matrix $C$ satisfying the following conditions:*

  - *The random variable $X$ conditioned on $Y = y$ is a continuous random variable with density functions $dP_{X|Y=y}$ and $dQ_{X|Y=y}$, respectively.*

  - *There exists $S \subseteq \mathcal{X}$ such that $P_X(S) + Q_X(S) > 0$, and $P_{X,Y}(x, \cdot) - Q_{X,Y}(x, \cdot)$ is not a right eigenvector of $C$, for all $x \in S$, where $P_{X,Y}(x, \cdot) = [P_{X,Y}(x, 1), \cdots, P_{X,Y}(x, m)]^T$.*

A pair $(P, Q)$ violating the above assumptions either has $X$ that is a mixture of continuous and discrete distribution, or all $(P(x, \cdot) - Q(x, \cdot))$'s are aligned with the right eigenvectors of $C$.

**Theorem 4.** *For all sufficiently small $\epsilon > 0$, all distributions $P_{X,Y}$ and $Q_{X,Y}$ satisfying Assumption 2, and all full-rank $C \in \mathbb{R}^{m \times m}$, there exist $\mathcal{F}_3$ not satisfying Assumption 1, such that*
$$d_{\mathcal{F}_3}(\widetilde{P}, \widetilde{Q}) = O_\epsilon(\epsilon) \quad and \quad d_{\mathcal{F}_3}(P, Q) = \Omega_\epsilon(1), \tag{25}$$
*and $\mathcal{F}_4$ not satisfying Assumption 1, such that*
$$d_{\mathcal{F}_4}(\widetilde{P}, \widetilde{Q}) = \Omega_\epsilon(1) \quad and \quad d_{\mathcal{F}_4}(P, Q) = O_\epsilon(\epsilon). \tag{26}$$

We refer to Appendix H for a proof. This implies that some assumptions on the function class $\mathcal{F}$ are necessary, such as those in Assumption 1. Without any restrictions, we can find bad examples where the two distances $d_\mathcal{F}(P, Q)$ and $d_\mathcal{F}(\widetilde{P}, \widetilde{Q})$ are arbitrarily different for any $C$, $P_{X,Y}$, and $Q_{X,Y}$.

# F Proof of Theorem 2

We start with the following two key lemmas.

**Lemma 4.** *For any sample $(X, Y) \sim P$ (or $Q$), let $\widetilde{P}$ (or $\widetilde{Q}$) denote the sample whose label $Y$ is corrupted by a noise defined by a confusion matrix $C$, where $C_{y\widetilde{y}}$ is the probability the a label $y$ is corrupted as a label $\widetilde{y}$. Then, for any $P$, $Q$, and any class of discriminators $\mathcal{F}$,*

$$d_{\mathcal{F}}(\widetilde{P}, \widetilde{Q}) \;=\; d_{C \circ \mathcal{F}}(P, Q) \,. \tag{27}$$

**Lemma 5.** *If a class of discriminators $\mathcal{F}$ satisfies Assumption 1 with a constant shift $c$, then for all row-stochastic non-negative matrix $C$,*

$$\|C^{-1}\|_{\infty}^{-1} \mathbf{I} \circ (\mathcal{F} - c) \;\subseteq\; C \circ (\mathcal{F} - c) \;\subseteq\; (\mathcal{F} - c) \,. \tag{28}$$

From the ordering of sets of functions in Lemma 5, it follows that

$$d_{C \circ \mathcal{F}}(P, Q) \;\geq\; d_{\|C^{-1}\|_{\infty}^{-1} \mathbf{I} \circ \mathcal{F}}(P, Q) \;=\; \|C^{-1}\|_{\infty}^{-1} d_{\mathcal{F}}(P, Q) \tag{29}$$

where the last equality follows from the fact that $D \in \mathcal{F}$ if and only if $\alpha D \in \alpha \mathbf{I} \circ \mathcal{F}$. Note that we ignored the shift $c$, as the definition of the neural network distance is invariant to any constant shift $c$, and we can cancel any given shift $c$ of a set $\mathcal{F}$ as we see fit. Next, it follows immediately from Lemma 5 that

$$d_{C \circ \mathcal{F}}(P, Q) \;\leq\; d_{\mathcal{F}}(P, Q) \,. \tag{30}$$

This finishes the proof of the theorem.

## F.1 Proof of Lemma 4

For any function $D : \mathcal{X} \times [m] \to \mathbb{R}$ and a distribution $P$ over $\mathcal{X} \times [m]$, we denote the expectation by $\langle D, P \rangle = \mathbb{E}_{(X,Y) \sim P}[D(X, Y)]$. Further, we let $\widetilde{P}(S, \widetilde{y}) = \sum_y P(S, y) C_{y\widetilde{y}} = PC(S, \widetilde{y})$ denote the distribution of the corrupted sample, by noise with confusion matrix $C$. Note that we intentionally overloaded the matrix multiplication notation $PC$. We also treat $D$ as a (infinite-dimensional) matrix, and let $DC^T(x, y) = \sum_{\widetilde{y}} D(x, \widetilde{y}) C_{y\widetilde{y}}$. It follows that,

$$
\begin{aligned}
d_{\mathcal{F}}(\widetilde{P}, \widetilde{Q}) \;&\triangleq\; \sup_{D \in \mathcal{F}} \; \mathbb{E}_{\widetilde{P}}[D(X, Y)] - \mathbb{E}_{\widetilde{Q}}[D(X, Y)] \\
&=\; \sup_{D \in \mathcal{F}} \langle D, \widetilde{P} - \widetilde{Q} \rangle \\
&=\; \sup_{D \in \mathcal{F}} \langle D, (P - Q)C \rangle \\
&=\; \sup_{D \in \mathcal{F}} \langle DC^T, (P - Q) \rangle \\
&=\; \sup_{\widetilde{D} \in C \circ \mathcal{F}} \langle \widetilde{D}, (P - Q) \rangle \,,
\end{aligned}
$$

where the last equality follows from the definition of the $\circ$ operation in Eq. (6). This proves the desired claim.

## F.2 Proof of Lemma 5

We will prove the desired claim for $\mathcal{F}_1$ and $\mathcal{F}_2$ separately. As the set orderings are preserved under addition of sets, this proves the desired claim.

First we will prove the Lemma when condition $T \circ \mathcal{F} \subseteq \mathcal{F}$ holds. We use the notations from AppendixF.1. We want to prove,

$$
\begin{aligned}
&\|C^{-1}\|_{\infty}^{-1} \mathbf{I} \circ \mathcal{F} \subseteq C \circ \mathcal{F} \\
\Longleftrightarrow\; &\forall D \in \mathcal{F}, \; \exists D' \in \mathcal{F} \text{ such that } \|C^{-1}\|_{\infty}^{-1} D(x, y) = D' C^T(x, y), \quad \forall x \in \mathcal{X}, \forall y \in [m] \\
\Longleftrightarrow\; &\forall D \in \mathcal{F}, \; \exists D' \in \mathcal{F} \text{ such that } \|C^{-1}\|_{\infty}^{-1} D C^{-T}(x, y) = D'(x, y), \quad \forall x \in \mathcal{X}, \forall y \in [m] \\
\Longleftrightarrow\; &\|C^{-1}\|_{\infty}^{-1} C^{-1} \circ \mathcal{F} \subseteq \mathcal{F} \,.
\end{aligned}
$$

The last statement is true by the the Assumption 1 because $\left\lVert\left(C^{-1}/\lVert\lvert C^{-1}\rvert\rVert_\infty\right)\right\rVert_\infty = 1$. The second covering $C \circ \mathcal{F} \subseteq \mathcal{F}$ is also true by Assumption 1 because $\lVert\lvert C\rvert\rVert_\infty = 1$ since $C$ is row-stochastic matrix with rows summing to 1.

Now we will prove the case when condition $\mathcal{F} = \{\alpha f(x) \mid f(x)_y = g(x) \in \mathbb{R}, g \in \mathcal{F}_x, \alpha \in [0,1]\}$ holds. $C \circ \mathcal{F} \subseteq \mathcal{F}$ holds true because $\mathbb{1}$ is an eigenvector of $C$ with eigenvalue 1. Similarly to the previous condition, we need to prove that $C^{-1}/\lVert\lvert C^{-1}\rvert\rVert_\infty \circ \mathcal{F} \subseteq \mathcal{F}$, but since $\mathbb{1}$ is an eigenvector of $C^{-1}$ with eigenvalue $1/\lVert\lvert C^{-1}\rvert\rVert_\infty \leq 1$, the again holds true.

# G    Proof of Remarks 2 and 1

## G.1    Class of all bounded functions (Total Variation)

Let $\mathcal{F}([c_1, c_2])$ be class of all functions with range inside $[c_1, c_2]^m$. Proof follows from the Appendix G.2 by taking $\lim_{L\to\infty}$.

## G.2    Class of all bounded and Lipschitz functions in $x$

Let $\mathcal{F}(L, [c_1, c_2])$ be class of all vector valued $L$-Lipschitz functions in $x$ with range inside $[c_1, c_2]^m$. That is,

$$\lVert D(x_1) - D(x_2)\rvert\rVert_\infty \leq L\lVert x_1 - x_2\rVert_2 \text{ and } D(x) \in [c_1, c_2] \ \forall x, x_1, x_2 \in \mathcal{X}, y \in [m] \tag{31}$$

**Lemma 6.** $d_{\mathcal{F}(L,[c_1,c_2])}(P, Q) = \frac{(c_2-c_1)}{2} \, d_{\mathcal{F}(2L/(c_2-c_1),[-1,1])}(P, Q)$

*Proof.* There exists a bijection $f : \mathcal{F}(L, [c_1, c_2]) \to \mathcal{F}(2L/(c_2 - c_1), [-1, 1])$, such that $f(D) = \frac{2D-(c_1+c_2)}{c_2-c_1}$. This is true since $f$ is invertible, $f(D) \in [-1, 1]$, and $f$ is $2L/(c_2 - c_1)$-Lipschitz.

$$
\begin{aligned}
d_{\mathcal{F}(L,[c_1,c_2])}(P, Q) &= \sup_{D\in\mathcal{F}(L,[c_1,c_2])} \ \mathbb{E}_{(x,y)\sim P}[D(x,y)] - \mathbb{E}_{(x,y)\sim Q}[D(x,y)] \\
&= \frac{(c_2-c_1)}{2} \sup_{D\in\mathcal{F}(2L/(c_2-c_1),[-1,1])} \ \mathbb{E}_{(x,y)\sim P}[f(D)(x,y)] - \mathbb{E}_{(x,y)\sim Q}[f(D)(x,y)] \\
&= \frac{(c_2-c_1)}{2} d_{\mathcal{F}(2L/(c_2-c_1),[-1,1])}(P, Q)
\end{aligned}
\tag{32}
$$

$\square$

By Lemma 6 any $[c_1, c_2]$ is similar to $[-1, 1]$ up to a scaling, thus we only prove for $[-1, 1]$. Now we will show that the inclusion condition (Assumption 1) hold for this class of functions.

**Lemma 7.** $T \circ \mathcal{F}(L, [-1, 1]) \subseteq \mathcal{F}(L, [-1, 1]), \forall \lVert\lvert T\rvert\rVert_\infty = 1$

*Proof.* We want to show that, $\forall \ D' \in T \circ \mathcal{F}(L, [-1, 1])$, we also have $D' \in \mathcal{F}(L, [-1, 1])$. In other words, $\forall \ D \in \mathcal{F}(L, [-1, 1]), TD \in \mathcal{F}(L, [-1, 1])$. First the we show that range of $D \in T \circ \mathcal{F}(L, [-1, 1])$ is $[-1, 1]$, i.e.,

$$\lVert TD(x)\rvert\rVert_\infty \leq \lVert\lvert T\rvert\rVert_\infty \lVert D(x)\rvert\rVert_\infty \leq 1 \cdot 1 \leq 1. \tag{33}$$

In a similar way we prove the Lipschitz property.

$$\lVert TD(x_1) - TD(x_2)\rvert\rVert_\infty \leq \lVert\lvert T\rvert\rVert_\infty \lVert D(x_1) - D(x_2)\rvert\rVert_\infty \leq 1 \cdot L\lVert x_1 - x_2\rVert_2 \tag{34}$$

$\square$

Finally, we can use Theorem 2 to get the desired result.

## G.3    Class of all bounded and Lipschitz functions in $x$ and $y$

As $|y_1 - y_2| \geq 1$ for all $y_1 \neq y_2$, $L$-Lipschitz functions with $L \geq c_2 - c_1$ only imposes conditions on pairs of data with the same values of $y$. Hence, this boils down to the previous case studied in Appendix G.2.

## G.4 Class of projection discriminators

The function class is $\mathcal{F} = \{f_1 + f_2 | f_1 \in \mathcal{F}_1^{(\theta)}, f_2 \in \mathcal{F}_2^{(\theta)}, \theta \in \mathbb{R}^{d_\theta}\}$, where

$$\mathcal{F}_1^{(\theta)} = \{\, \text{vec}(y)^T V \, \psi(x;\theta) \,|\, V \in \mathcal{V}_1 \,\}, \text{ and,} \tag{35}$$

$$\mathcal{F}_2^{(\theta)} = \{\, v^T \, \psi'(x;\theta) \,|\, v \in \mathcal{V}_2 \,\}, \text{ where,} \tag{36}$$

$$\mathcal{V}_1 = \{\, V \in \mathbb{R}^{m \times d_V} \,|\, \max_i |V_{ij}| \le 1 \text{ for all } j \in [d_V] \,\}, \text{ and} \tag{37}$$

$$\mathcal{V}_2 = \{\, v \in \mathbb{R}^{d_v} \,|\, \|v\| \le 1 \,\}. \tag{38}$$

We will show that both $\mathcal{F}_1^{(\theta)}$ and $\mathcal{F}_2^{(\theta)}$ satisfy Assumption 1. For any $A \in \mathbb{R}^{m \times m}$, we can write $A \circ \mathcal{F}_1^{(\theta)}$ as

$$A \circ \mathcal{F}_1^{(\theta)} = \{\text{vec}(y)^T AV \psi(x;\theta) \,|\, V \in \mathcal{V}_1 \}. \tag{39}$$

If $\|\|A\|\|_\infty = 1$, then, $\max_{i \in [m]} |(AV)_{ij}| = \|(AV)_{.,j}\|_\infty = \|A(V_{.,j})\|_\infty \le \|\|A\|\|_\infty \|\|V_{.,j}\|\|_\infty = 1 \cdot \max_{i \in [m]} |V_{ij}| \le 1$, which implies than $AV \in \mathcal{V}_1$. Thus $\mathcal{F}_1^{(\theta)}$ satisfies inclusion condition, $A \circ \mathcal{F}_1^{(\theta)} \subseteq \mathcal{F}_1^{(\theta)}$. Since

$$\mathcal{V}_2 = \{\, v \in \mathbb{R}^{d_v} \,|\, \|v\| \le 1 \,\} = \{\, \alpha v \,|\, v \in \mathbb{R}^{d_v}, \|v\| = 1, \alpha \in [0,1] \,\} \tag{40}$$

we can re-write $\mathcal{F}_2^{(\theta)}$ as,

$$\mathcal{F}_2^{(\theta)} = \left\{ \alpha g(x,y) \mid g(x,y) = f(x), \text{for any } f(x) \in \{v^T \psi'(x;\theta) \,|\, \|v\| = 1\}, \text{ and } \alpha \in [0,1] \right\}. \tag{41}$$

Thus $\mathcal{F}_2^{(\theta)}$ satisfies the label invariance condition. Finally, since Assumption 1 holds true for $\mathcal{F}_1^{(\theta)}$ and $\mathcal{F}_2^{(\theta)}$, it also holds for $\mathcal{F}$.

## H Proof of Theorem 4

Let $\mathcal{F} = \{D \,|\, D(x) \in [-1,1]^m\}$. We show that,

$$\mathcal{F}_3 = \left\{ D \in \mathcal{F} \,\middle|\, \left|D(x)^T C^T (\bar{p}(x) - \bar{q}(x))\right| \le \epsilon \; \forall x \in \mathcal{X} \right\}.$$

$$\mathcal{F}_4 = \left\{ D \in \mathcal{F} \,\middle|\, \left|D(x)^T (\bar{p}(x) - \bar{q}(x))\right| \le \epsilon \; \forall x \in \mathcal{X} \right\}$$

Then,

$$
\begin{aligned}
d_{\mathcal{F}_3}(\widetilde{P}, \widetilde{Q}) &= \sup_{D \in \mathcal{F}} \mathbb{E}_{(x,y)\sim\widetilde{P}}[D(x)_y] - \mathbb{E}_{(x,y)\sim\widetilde{Q}}[D(x)_y] \\
&= \sup_{D \in \mathcal{F}} \mathbb{E}_{(x,y)\sim P}[(CD(x))_y] - \mathbb{E}_{(x,y)\sim Q}[(CD(x))_y] \\
&= \sup_{D \in \mathcal{F}} \int_{\mathcal{X}} D(x)^T C^T (p(x)P_{Y|X=x}(y)) - q(x)Q_{Y|X=x}(y)) \\
&\leq \sup_{D \in \mathcal{F}} \int_{\mathcal{X}} \epsilon = O(\epsilon) \\
d_{\mathcal{F}_3}(P, Q) &= \sup_{D \in \mathcal{F}} \mathbb{E}_{(x,y)\sim P}[D(x)_y] - \mathbb{E}_{(x,y)\sim Q}[D(x)_y] \\
&= \sup_{D \in \mathcal{F}} \int_{\mathcal{X}} D(x)^T (p(x)\overline{P}_{Y|X=x} - q(x)\overline{Q}_{Y|X=x}) \\
&\geq \sup_{D \in \mathcal{F}} \int_{\mathcal{X}_S} D(x)^T (p(x)\overline{P}_{Y|X=x} - q(x)\overline{Q}_{Y|X=x}) \\
&\overset{(a)}{\geq} \sup_{\substack{D \in \mathcal{F} \\ D(x) \perp v_x}} \int_{\mathcal{X}_S} D(x)^T u_x \\
&= \sup_{D \in \mathcal{F}} \int_{\mathcal{X}_S} D(x)^T (u_x - (u_x^T \hat{v}_x)\hat{v}_x) \\
&= \sup_{D \in \mathcal{F}} \int_{\mathcal{X}_S} \|\|u_x - (u_x^T \hat{v}_x)\hat{v}_x)\|\|_1,
\end{aligned}
$$

where in $(a)$ we put $p(x)P_{Y|X=x}(y) - q(x)Q_{Y|X=x}(y) = u_x$ and $C^T u_x = v_x$. Since $u_x$ is not and eigenvector of $C$, $u_x \not\perp v_x$ and therefore the integrand is positive. Finally using the assumption that $P_X(\mathcal{X}_S) + Q_X(\mathcal{X}) > 0$ we get that LHS is positive number. Now by taking $\epsilon$ much smaller than the LHS we get the desired result. Other case also follows similarly.

# I  Proof of Theorem 3

**Proposition 1.** *There exists a class $\mathcal{F}$ of parametric vector valued functions which satisfy the Lipschitzness in parameters property* (12) *and such that $\mathcal{F} - 1/2\mathbb{1}$ satisfy inclusion condition (Assumption 1).*

*Proof.* For the proof we show that there exists a class of discriminators which satisfy the inclusion condition of Assumption 1 and in particular we have the following example. Let $\mathcal{F}'$ be a class of vector functions parameterized by the $u' \in \mathcal{U}'$ which is $L'$-Lipschitz in the parameters and who element functions satisfy $\|\|f_{u'}(x)\|\|_\infty \leq 1/2$. We define a new class of vector functions

$$
\overline{\mathcal{F}} \triangleq \{T'f_{u'}(\cdot) + 1/2\,\mathbb{1} \mid \forall f(\cdot) \in \mathcal{F}', \|\|T'\|\|_\infty \leq 1\}, \tag{42}
$$

parameterized by $u = (u', T') \in \mathcal{U}' \times \{T' \mid \|\|T'\|\|_\infty \leq 1\}$. Then,

$$
\begin{aligned}
\|\|T'_1 D_{u'_1}(x, \cdot) - T'_2 D_{u'_2}(x, \cdot)\|\|_\infty &\overset{(a)}{\leq} \|\|T'_1 D_{u'_1}(x, \cdot) - T'_2 D_{u'_1}(x, \cdot)\|\|_\infty + \|\|T'_2 D_{u'_1}(x, \cdot) - T'_2 D_{u'_2}(x, \cdot)\|\|_\infty \\
&\overset{(b)}{\leq} \|\|T'_1 D_{u'_1}(x, \cdot) - T'_2 D_{u'_1}(x, \cdot)\|\|_2 + \|\|T'_2\|\|_\infty \|\|D_{u'_1}(x, \cdot) - D_{u'_2}(x, \cdot)\|\|_\infty \\
&\overset{(c)}{\leq} \|\|T'_1 - T'_2\|\|_2 \|\|D_{u'_1}(x, \cdot)\|\|_2 + \|\|T'_2\|\|_\infty L' \|\|u'_1 - u'_2\|\|_2 \\
&\overset{(d)}{\leq} \|\|T'_1 - T'_2\|\|_2 \sqrt{m} + L' \|\|u'_1 - u'_2\|\|_2 \\
&\leq (\sqrt{m} + L') (\|\|T'_1 - T'_2\|\|_2 + \|\|u'_1 - u'_2\|\|_2) \\
&\overset{(e)}{\leq} \sqrt{2}(\sqrt{m} + L') \sqrt{\|\|T'_1 - T'_2\|\|_2^2 + \|\|u'_1 - u'_2\|\|_2^2}
\end{aligned} \tag{43}
$$

where $(a)$ uses triangle inequality, $(b)$ and $(c)$ uses $\|\|x\|\|_\infty \leq \|\|x\|\|_2$ and $\|Ax\|_p \leq \|A\|_p \|x\|_p$ (see Section 1), $(c)$ is true since $\overline{\mathcal{F}}'$ is $L'$-Lipschitz in $u'$, $(d)$ uses $\|\|D_{u'}(x,\cdot)\|\|_\infty \leq 1/2$ and $\|\|T_2'\|\|_\infty \leq 1$, and $(e)$ uses $x + y \leq \sqrt{2(x^2 + y^2)}$.

Next we show that $\mathcal{F}$ lies in the range $[0,1]^m$ as follows.

$$\|\|T'D_{u'}(x,\cdot)\|\|_\infty \leq \|\|T'\|\|_\infty \|\|D_{u'}(x,\cdot)\|\|_\infty \overset{(b)}{\leq} 1 \cdot 1/2 \tag{44}$$

We can prove inclusion condition in Assumption 1 by the fact that $\|\|TT'\|\|_\infty \leq \|\|T\|\|_\infty \|\|T'\|\|_\infty \leq 1 \cdot 1$ and hence $TT'$ is valid choice for for $T$. $\square$

Next we present a straightforward corollary of Theorem 2.

**Theorem 5.** *Let $\mathcal{F}$ be a parametric class of vector valued functions parameterized by $u \in \mathcal{U} \subseteq \mathbb{R}^p$ such that $f_u : \mathcal{X} \to [0,1]^m$, $\forall f_u \in \mathcal{F}$. Further, if $\mathcal{F}$ is $L$-Lipschitz in the parameter $u$, as defined in equation (12), and if $\mathcal{F}$ satisfies Assumption 1, then,*

$$d_{\mathcal{F}}(\widetilde{P},\widetilde{Q}) \leq d_{\mathcal{F}}(P,Q) \leq \|\|C^{-1}\|\|_\infty d_{\mathcal{F}}(\widetilde{P},\widetilde{Q}) \tag{45}$$

*Proof.* A proof directly follows from Theorem 2. $\square$

**Corollary 1** (of [3], Theorem 3.1.)**.** *Assume the same class $\mathcal{F}$ as in Theorem 5. Let $\widetilde{P}$, $\widetilde{Q}$ be two distributions on $X$, $\widetilde{Y}$ and be $\widetilde{P}_n$, $\widetilde{Q}_n$ be empirical versions of them with at least $n$ samples each. Then there is universal constant $c$ such that when $n \geq \frac{cp\log(pL/\epsilon)}{\epsilon^2}$, we have with probability at least $1 - exp(-p)$ over the randomness of $\widetilde{P}_n$, $\widetilde{Q}_n$,*

$$\left| d_{\mathcal{F}}(\widetilde{P}_n,\widetilde{Q}_n) - d_{\mathcal{F}}(\widetilde{P},\widetilde{Q}) \right| \leq \epsilon \tag{46}$$

*Proof.* Proof is directly follow from [3][Theorem 3.1.]. $\square$

$$d_{\mathcal{F}}(\widetilde{P},\widetilde{Q}) \overset{(a)}{\leq} d_{\mathcal{F}}(P,Q) \overset{(a)}{\leq} \|\|C^{-1}\|\|_\infty d_{\mathcal{F}}(\widetilde{P},\widetilde{Q})$$

$$d_{\mathcal{F}}(\widetilde{P}_n,\widetilde{Q}_n) - \epsilon \overset{(a)}{\leq} d_{\mathcal{F}}(P,Q) \overset{(a)}{\leq} \|\|C^{-1}\|\|_\infty (d_{\mathcal{F}}(\widetilde{P}_n,\widetilde{Q}_n) + \epsilon)$$

where $(a)$ is true by Theorem 5 and $(b)$ is true from the Corollary 1.

## J   Numerical comparisons to AmbientGAN [10]

In Table 1, we plot the generated label accuracy (as defined in Section 5.1) of RCGAN (which uses the proposed projection discriminator) and AmbientGAN (which uses the DCGAN with no projection discriminator) for multiple values of noise levels $(1 - \alpha)$. One of the main reasons for the performance drop of AmbientGAN is that without the projection discriminator, training of AmbientGAN is sensitive to how the mini-batches are chosen. For example, if the distribution of the labels in the mini-batch of the real data is different from that of the mini-batch of the generated data, then the performance of (conditional) AmbientGAN significantly drops. This is critical as we have noisy labels, and matching the labels is in the mini-batch is challenging. Our proposed RCGAN provides an architecture and training methods for applying AmbientGAN to noisy labeled data, to overcome theses challenges. When a projection discriminator is used, as in all our RCGAN and RCGAN-U implementations, the performance is not sensitive to how the mini-batches are sampled. When a discriminator that is not necessarily a projection discriminator is used, as in our RCGAN+$y$ architecture, we propose a novel scheduling of the training, which avoids local minima resulting from mis-matched mini-batches (explained in Appendix L). The results are averaged over 10 instances.

|  | Noise level (1-$\alpha$) | | |
|---|---|---|---|
|  | 0.2 | 0.3 | 0.5 |
| **RCGAN** | 0.994 | 0.994 | 0.994 |
| **AmbientGAN** | 0.940 | 0.902 | 0.857 |

Table 1: Noisy MNIST dataset: in generated label accuracy, RCGAN improves upon the standard implementation of the AmbientGAN with DCGAN architecture (we refer to Appendix L for implementation details).

## K   Mixing regularizer for RCGAN

In addition to the permutation regularizer, we also propose another novel regularizer; the *mixing regularizer* (controlled by $\lambda'$). Mixing error happens if, when generator is asked to produce samples from one class, it produces a mix of different classes. To combat this, we propose the mixing regularizer, which tries to minimize the best possible classification loss $\mathbb{E}\left[\ell(h_2(x), y)\right]$ when asked to predict $Y$ given $X$. This encourages the generated samples from different classes to be different. Maximizing the mixing regularizer is similar to maximizing the InfoGAN loss [11], which is a variational lower bound on the mutual information between $Y$ and $X$.

Figure 4: The output $x$ of the conditional generator $G$ is paired with a noisy label $\widetilde{y}$ corrupted by the channel $C$. The discriminator $D$ estimates whether a given labeled sample is coming from the real data $(x_{\text{real}}, \tilde{y}_{\text{real}})$ or generated data $(x, \tilde{y})$. The permutation regularizer $h_1$ is pre-trained on real data.

With this regularizer, the D-step and G-step of the RCGAN(-U) are modified as follows. We train on the following loss with $\lambda, \lambda' > 0$.
**D-step:**

$$\max_{D \in \mathcal{F}, h_2 \in \mathcal{H}} \mathbb{E}_{(x, \widetilde{y}) \sim \widetilde{P}_{X, \widetilde{Y}}} \left[\phi\left(D(x, \widetilde{y})\right)\right] + \mathbb{E}_{\substack{z \sim N \, y \sim P_Y \\ \widetilde{y}|y \sim C_y}} \left[\phi\left(1 - D(G(z; y), \widetilde{y})\right) - \lambda' l(h_2(G(z; y), y)\right].$$

**G-step:**

$$\min_{G \in \mathcal{G}} \mathbb{E}_{\substack{z \sim N \, y \sim P_Y \\ \widetilde{y}|y \sim C_y}} \left[\phi\left(1 - D(G(z; y), \widetilde{y})\right) + \underbrace{\lambda \, l(h_1^*(G(z; y)), y)}_{\text{permutation regularizer}} + \underbrace{\lambda' \, l(h_2(G(z; y)), y)}_{\text{mixing regularizer}}\right]. \qquad (47)$$

However, in our experiments, we did not see any performance gain when using the mixing regularizer. This is possibly because the generator architecture naturally enforces the separation of classes.

## L   Implementation details

**Hyper-parameters and architectures for MNIST:** Biased GAN uses the standard conditional DCGAN architecture [37] implementation[2]. RCGAN, RCGAN-U, and unbiased GAN use the same DCGAN architecture [37] with hinge loss, $\phi(a) = \max(0, 1 - 2a)$, and conditional projection discriminator [32], as suggested by our theoretical analysis. Additionally, we also present RCGAN+y

architecture, which has the same architecture as RCGAN but the input to the first layer of its discriminator is concatenated with a one-hot representation of the label.

We use $\lambda = 0$ and $\lambda = 1$ for RCGAN and RCGAN-U respectively, and a linear classifier for the permutation regularizer. For RCGAN-U, the learning rate of the confusion matrix is 10 times as that of the discriminator and the generator.

**RCGAN+y training:** RCGAN+y architecture has the same architecture as RCGAN but the input to the first layer of its discriminator is concatenated with a one-hot representation of the label. We observed that RCGAN+y is harder to train especially at low noise regimes of $\alpha \geq 0.4$. To combat this we add additional artificial noise, parameterized by $\tilde{\alpha}$, so that the effective noise parameter is $\bar{\alpha} = \tilde{\alpha}(\alpha - \frac{(1-\alpha)}{9}) + \frac{(1-\alpha)}{9}$. We schedule $\tilde{\alpha}$ during the training so that from epoch 0 to 30, the effective noise $\bar{\alpha} = 0.3$ and from epoch 30 to 80, we linearly reduce $\tilde{\alpha}$ so that the effective noise linearly decrease from 0.3 to $\alpha$. Finally from epoch 80 to 100 we keep the artificial noise parameter $\tilde{\alpha}$ to 1, so that effective noise $\bar{\alpha} = \alpha$. This stabilizes the training of RCGAN+y.

We suspect that the reason why RCGAN+y works better than RCGAN is that RCGAN+y optimizes over a larger class of functions, which might be necessary when learning a conditional distribution with large noise. Thus, as the noise increases it becomes more challenging for the projection discriminator to differentiate between $\widetilde{P}$ and $\widetilde{Q}$, since it is much simpler function of $y$ with $y$ appearing only in the final layer. However, in RCGAN+y since we concatenate $y$ to the input of the first layer, RCGAN+y discriminator may be able to better differentiate between $\widetilde{P}$ and $\widetilde{Q}$.

**Hyper-parameters and architectures for CIFAR-**10**:** We use ResNet based GAN used in [15] with spectral normalization [31] and conditional projection discriminator [32] [3]. We note that the spectral normalization work [31] reports higher inception score than what we achieved on the noiseless setting, possibly due to limited hyper-parameter tuning. For all the four approaches we use the same hyper-parameters.

We use $\lambda = 0$ and $\lambda = 1$ for RCGAN and RCGAN-U respectively, and a linear classifier for the permutation regularizer. For RCGAN-U, the learning rate of the confusion matrix is same as that of the discriminator and the generator.

**RCGAN-U CIFAR-**10 **initialization:** For CIFAR-10 dataset, we observed that even with the permutation regularizer, the learned confusion matrix in RCGAN-U was a permuted version of the true $C$, possibly because a linear classifier might be too simple to classify CIFAR images. To combat this, we initialized the confusion matrix $M$ to be diagonally dominant. We initialized the confusion matrix to be learned $M$ so that diagonal entries are 0.2 and off-diagonal entries are $(1 - 0.2)/9$. In our experiments, this ensured that the approximately true confusion matrix $C$ was learned by $M$ in RCGAN-U. We believe that better CIFAR-10 classifier for permutation regularizers can achieve the same effect as this initialization.

# M  Additional experimental information

In this section, we provide tables with the numerical values of the data points in the Figures 2 and 3.

| Noise $(1 - \alpha)$ | RCGAN+y | RCGAN | RCGAN-U | unbiased GAN | biased GAN |
|---|---|---|---|---|---|
| 0.875 | 0.905 | 0.11 | 0.215 | 0.1 | 0.119 |
| 0.85 | 0.984 | 0.211 | 0.235 | 0.1 | 0.138 |
| 0.8 | 0.987 | 0.44 | 0.489 | 0.1 | 0.192 |
| 0.7 | 0.985 | 0.978 | 0.992 | 0.1 | 0.288 |
| 0.6 | 0.976 | 0.983 | 0.991 | 0.2 | 0.375 |
| 0.5 | 0.976 | 0.991 | 0.986 | 0.869 | 0.475 |
| 0.4 | 0.984 | 0.994 | 0.99 | 0.997 | 0.57 |
| 0.3 | 0.983 | 0.994 | 0.987 | 0.991 | 0.681 |
| 0.2 | 0.99 | 0.994 | 0.995 | 0.999 | 0.765 |
| 0.1 | 0.99 | 0.994 | 0.995 | 0.998 | 0.873 |
| 0.0 | 0.995 | 0.995 | 0.994 | 0.994 | 0.994 |

Table 2: Numerical values for the data-points in Figure 2 (left panel). Noisy MNIST dataset: Generated label accuracy of RCGAN, RCGAN-U, RCGAN+y, unbiased GAN and biased GAN.

| Noise $(1 - \alpha)$ | RCGAN+y | RCGAN | RCGAN-U | unbiased GAN | biased GAN |
|---|---|---|---|---|---|
| 0.875 | 0.156 | 0.885 | 0.86 | 0.898 | 0.872 |
| 0.85 | 0.102 | 0.774 | 0.77 | 0.894 | 0.854 |
| 0.8 | 0.088 | 0.638 | 0.69 | 0.9 | 0.634 |
| 0.7 | 0.11 | 0.096 | 0.098 | 0.768 | 0.55 |
| 0.6 | 0.088 | 0.1 | 0.058 | 0.902 | 0.322 |
| 0.5 | 0.07 | 0.106 | 0.094 | 0.472 | 0.274 |
| 0.4 | 0.072 | 0.098 | 0.08 | 0.158 | 0.164 |
| 0.3 | 0.096 | 0.088 | 0.084 | 0.098 | 0.142 |
| 0.2 | 0.076 | 0.086 | 0.086 | 0.07 | 0.138 |
| 0.1 | 0.112 | 0.068 | 0.096 | 0.088 | 0.104 |
| 0.0 | 0.069 | 0.069 | 0.069 | 0.069 | 0.069 |

Table 3: Numerical values for the data-points in Figure 2 (right panel). Noisy MNIST dataset: Label recovery error of RCGAN, RCGAN-U, RCGAN+y, unbiased GAN and biased GAN.

| Noise $(1 - \alpha)$ | RCGAN | RCGAN-U | unbiased GAN | biased GAN |
|---|---|---|---|---|
| 0.8 | 0.111 | 0.126 | 0.110 | 0.117 |
| 0.6 | 0.443 | 0.263 | 0.1 | 0.148 |
| 0.4 | 0.700 | 0.71 | 0.4 | 0.340 |
| 0.2 | 0.724 | 0.815 | 0.6 | 0.507 |
| 0.0 | 0.751 | 0.751 | 0.751 | 0.751 |

Table 4: Numerical values for the data-points in Figure 3 (left panel). Noisy CIFAR-10 dataset: Generated label accuracy of RCGAN, RCGAN-U, unbiased GAN and biased GAN.

| Noise $(1 - \alpha)$ | RCGAN | RCGAN-U | unbiased GAN | biased GAN |
|---|---|---|---|---|
| 0.8 | 7.8 | 7.58 | 4.37 | 7.6 |
| 0.6 | 7.85 | 7.56 | 4 | 7.68 |
| 0.4 | 8.05 | 8.03 | 4 | 7.75 |
| 0.2 | 8.11 | 8.12 | 6 | 7.91 |
| 0.0 | 8.13 | 8.13 | 8.13 | 8.13 |

Table 5: Numerical values for the data-points in Figure 3 (right panel). Noisy CIFAR-10 dataset: Inception score of RCGAN, RCGAN-U, unbiased GAN and biased GAN.

Figure 5: Images generated from the RCGAN+y, RCGAN , RCGAN-U, Unbiased and Biased GANs trained using noisy MNIST dataset, where class labels are flipped uniformly at random with probability 1 − 'real label accuracy', under the uniform flipping model.

Figure 6: Images generated from the RCGAN-U, Unbiased and Biased GANs trained using noisy CIFAR-10 dataset, where class labels are flipped uniformly at random with probability 1 − 'real label accuracy', under the uniform flipping model.

## Footnotes

[2]`https://github.com/carpedm20/DCGAN-tensorflow`

[3]`https://github.com/watsonyanghx/GAN_Lib_Tensorflow`