[Reviews · NeurIPS 2018]

Reviewer 1



This paper proposes to deal with the issue of noisy labels in training conditional GANs. In the first case, they consider the situation where the confusion matrix between the true label distribution and the perturbed label distribution (marginally) is known. In the second case, they consider the situation where this confusion matrix is not known. In their "RCGAN" model, two classifiers are trained. h1* is trained on real data samples with the noisy labels, and h2 is trained on generated samples to recover the label used to generate, and the generator is trained to produce samples conditioned on a label y which are classified to have the label y under both of these classifiers. In the training of the discriminator, for "real" samples the real x and the noisy corresponding label are both given to the discriminator. The generation process starts by sampling from the clean marginal distribution, running the generator conditioned on that clean label and providing that along with the corrupted label to the discriminator. Finally they show that by training this model it's possible to do an optimization problem over z in G(z,y) to assign labels to data points for which a number of noisy labels are available, as is the case in crowdsourcing. This seems like an elegant and very reasonable solution for an important problem. Small issues: -Calling the classifier h_1 is a bit confusing as h usually refers to hidden states. -In equation 1, I think the sample from the noisy y distribution should be called y_tilde and not y.

Reviewer 2



Conditional generative adversarial network (GAN) learn conditional distribution from a joint probability distribution but corrupted labels hinder this learning process. The paper highlights a unique algorithm to learn this conditional distribution with corrupted labels. The paper introduces two architectures i) RCGAN which relies on availability of matrix C which contains information regarding the errors, and ii) RCGAN-U which does not contain any information about the matrix C. The paper claims that there is no significant loss in performance with regards to knowledge about the matrix C. Even though the problem is unique in nature the paper contains details of some of the related work and references to techniques utilized in the paper such as projection discriminator. I believe the in-depth analysis of the assumptions with theorems and proofs solidify the claims made in the paper although the math requires a more careful check. The paper provides experimentation results on two separate datasets As far as I understand the permutation regularizer penalizes the model if generator produces samples from a class different from true one and mixing regularizer penalizes if it produces a mix of classes. The paper states that in practice we require a very weak classifier, an explanation of the intuition behind this or theoretical analysis of this statement should help concrete this claim. The paper utilizes recently proposed neural network distance which only guarantees some weak notions of generalization. Ambient GAN as stated in the related work section most closely related to the paper but the paper lacks a comparison with it under the experiments section. To summarize the paper highlights a unique problem. It would be interesting to see its application to fix corrupted labels in the training data.

Reviewer 3



Update: The authors do a good job of answering my questions. So I am raising my score. ****** This paper describes a method for learning a conditional GAN when there is significant corruption or noise in the labels. The paper proposes additional regularizers for training the generator and discriminator to achieve this. These regularizers that resemble standard auxiliary conditional GAN involve learning two additional classifiers. The first one is called the permutation regularizer and aims to reduce the mismatch between the generated label and true label. The other regularizer seems to encourage high confidence label predictions. How does the later relate to conditional entropy minimization [1]? 

The experiments are based on CIFAR10 and MNIST, showing the benefits of RCGAN over the original conditional-GAN. The discussions on the experiments could be expanded. Some of the image generation plots can be brought into the main paper for completeness.    It will also be nice to have some comparative results on the related Ambient GAN method. It is not clear what the authors mean by ‘sharp analysis for both the population loss and the finite sample loss’. 
 The organization and writing of the paper needs to improve a bit. Line 38 needs to be reworked. At several places symbols are used much before defining them. The theoretical results seems disconnected from the main text body, it will be useful to make them crisper and move less important details can be to the appendix. [1] Miyato, Takeru, et al. "Virtual adversarial training: a regularization method for supervised and semi-supervised learning." arXiv preprint arXiv:1704.03976 (2017).